# A Generalist Hanabi Agent

**Arjun Vaithilingam Sudhakar**[*,1,2,3] **Hadi Nekoei**[*,1,2,4] **Mathieu Reymond**[1,2,3]
**Janarthanan Rajendran**[6] **Miao Liu**[5] **Sarath Chandar**[1,2,3,7]

[1]Chandar Research Lab   [2]Mila – Quebec AI Institute   [3]Polytechnique Montréal
[4]Université de Montréal   [5]IBM Research   [6]Dalhousie University   [7]Canada CIFAR AI Chair

## Abstract

Traditional multi-agent reinforcement learning (MARL) systems can develop co-operative strategies through repeated interactions. However, these systems are unable to perform well on any other setting than the one they have been trained on, and struggle to successfully cooperate with unfamiliar collaborators. This is particularly visible in the Hanabi benchmark, a popular 2-to-5 player coopera-tive card-game which requires complex reasoning and precise assistance to other agents. Current MARL agents for Hanabi can only learn one specific game-setting (e.g., 2-player games), and play with the same algorithmic agents. This is in stark contrast to humans, who can quickly adjust their strategies to work with unfamiliar partners or situations. In this paper, we introduce Recurrent Replay Relevance Distributed DQN (R3D2), a generalist agent for Hanabi, designed to overcome these limitations. We reformulate the task using text, as language has been shown to improve transfer. We then propose a distributed MARL algorithm that copes with the resulting dynamic observation- and action-space. In doing so, our agent is the first that can play all game settings concurrently, and extend strategies learned from one setting to other ones. As a consequence, our agent also demonstrates the ability to collaborate with different algorithmic agents — agents that are themselves unable to do so. The implementation code is available at: R3D2-A-Generalist-Hanabi-Agent

## 1 Introduction

Humans were able to thrive as a society through their ability to cooperate. Interactions among multi-ple people or agents are essential components of various aspects of our lives, ranging from everyday activities like commuting to work, to the functioning of fundamental institutions like governments and economic markets. Through repeated interactions, humans can understand their partners, and learn to reason from their perspective. Crucially, humans can generalize their reasonings towards novel partners, in different situations. Artificial agents should be able to do the same for the suc-cessful collaboration of artificial and hybrid systems (Dafoe et al., 2020). This is why defining the problem of multi-agent cooperation nicely fits the multi-agent reinforcement learning (MARL) paradigm, as artificial agents learn to collaborate together through repeated interactions, in the same principled manner humans would.

In MARL, the game of Hanabi has emerged as a popular benchmark to assess the cooperative abil-ities of learning agents (Bard et al., 2020). Hanabi is a partially-observable card game designed for 2 to 5 players, with approximately $2^{90}$ unique player hands in the 5-player setting. Progressing in the game requires intricate skills, including long-term planning, precise assistance through clues to other agents, and complex reasoning. Adding to the complexity, players are required to infer the be-liefs and intentions of their counterparts through theory of mind reasoning (Bard et al., 2020). All of these characteristics are required in real-world multi-agent interactions, making Hanabi a challeng-ing and relevant testbed for MARL. Moreover, Hu et al. (2020); Hu & Sadigh (2023) have shown that agents performing well in Hanabi, particularly in zero-shot coordination, demonstrate improved capabilities in human-AI collaborative scenarios.

---

[*]Equal contribution. Corresponding authors: `arjun.vaithilingam-sudhakar@mila.quebec`, `nekoeihe@mila.quebec`

A straightforward way to learn to play is through self-play (Tan, 1993; Tampuu et al., 2017; Foerster et al., 2019). By repeatedly playing with oneself, an agent can learn conventions and effectively play the game. Sadly, these conventions often do not apply to others, resulting in misunderstandings and thus a drop in cooperation capabilities when paired with novel agents (Carroll et al., 2019). This is of course undesired behavior. An important aspect of assessing an artificial agent's performance is thus to evaluate its cooperative abilities when paired with agent it has not trained with, i.e., zero-shot coordination (ZSC). This means agents require to have a solid understanding of the game, and need robust strategies that cope with unexpected decisions of their teammates. Moreover, if a strategy is robust enough, it should provide a solid basis for variations of the task at hand. In Hanabi, for example, the optimal strategy for a 3-player game is different from the one for a 2-player game, even though the rules remain unchanged. However, using a robust 2-player strategy on a 3-player game should still yield solid performance, even if not optimal.

Learning such robust strategies is precisely the goal of this paper. By sacrificing some of the performance gains that come with learning a highly specialized but inflexible strategy, we design an agent that can not only generalize across different types of partners, but also to different game settings.

A centerpiece of our work lies in the realisation that the representation of the Hanabi environment the agents use to make decisions is highly structured, and thus inflexible. Changing the game setting results in a completely different structure, severely hampering the potential transfer of knowledge from one setting to another. This is the case for both observations of the game's state – an abstract encoding of bits – and actions the agent perform – a one-hot encoding for all action-combinations. Thus, as a first contribution, we modify the representations of both the observation- and action-spaces of Hanabi to make it more suitable for knowledge transfer. For this, we propose to use natural language as a backbone for our representation. Language has been shown to be a successful medium for transfer (Radford et al., 2019b; Brown et al., 2020), and using text for observations and actions results in a representation that becomes agnostic to number of partners in play (i.e., the setting of the game). This results in agents that learn on similar data-distributions, regardless of the number of teammates in play.

Next, we propose a novel neural network architecture that combines language models (Devlin et al., 2018b) and Deep Recurrent Relevance Q-network (DRRN) (He et al., 2015) to create an agent robust towards the dynamic textual observation- and action-spaces. Integrating this architecture with a distributed training regimen results in the Recurrent Replay Relevance Distributed DQN (R3D2) algorithm, our main contribution. What is remarkable is that, even though R3D2 learns the game of Hanabi through self-play, the simple fact of using a more abstract game representation and a well-suited network architecture results in robust strategies that to not only successfully cooperate with unseen R3D2 agents, but also – and perhaps more importantly – with completely different algorithmic agents. Moreover, due to their dynamic network architecture, R3D2 agents that have been trained on different player settings are able to collaborate together, even though they have learned different strategies.

Finally, because of the player-agnostic nature of R3D2, R3D2 agents can change the number of players in a game *while they are learning*, effectively enabling what we call *variable-player learning*, a multi-agent-specific variant of multi-task learning. By training with multiple combinations of number of agents, R3D2 can extend the simpler 2-player strategies to the hardest 5-player setting. In doing so, we have developed the first generalist Hanabi agent. To the best of our knowledge, this is the first time that generalization across game settings has been investigated for Hanabi. We argue this to be an essential component that defines robustness of behavior, and believe that generalization across game-settings is a promising research direction to evaluate policy robustness. While we demonstrate our approach on Hanabi, our core technical contributions - text-based representation for better transfer, architecture for dynamic action/state spaces, and variable-player learning are domain-agnostic advances that could benefit MARL applications in general.

## 2 RELATED WORK

**MARL for Hanabi** For successful collaboration, MARL agents require specific skills, such as dealing with imperfect information, predicting the intentions of partners, and communicating valuable information to others. All of these skills are required to play the game of Hanabi, which is why Bard et al. (2020) proposed the Hanabi challenge as a new frontier for AI research. The first

deep RL methods to learn winning strategies for Hanabi use self-play, combined with either a public belief state (Foerster et al., 2019), or by explicitly providing additional information during training (Hu & Foerster, 2019). However, as observed by Hu et al. (2020), self-play agents learn highly specialized conventions that are not transferable to novel partners. They introduce the concept of zero-shot coordination (ZSC), where AI agents need to coordinate with partners they have never seen before, and propose *Other-Play* for ZSC. Other-Play develops more robust strategies by leveraging the presence of known symmetries in the underlying problem. However, the ZSC performance is evaluated through *cross-play*, i.e., agents of the same learning algorithm but from independent training runs. While it is an important step towards generalization, cross-play is a cheap proxy to ZSC with completely different AI agents or even human players. Multiple works have since proposed ZSC-capable learners. Lupu et al. (2021); Nekoei et al. (2021) use population training with diverse policies so agents train with a large pool of agents, thus encountering diverse behaviors and creating robust strategies that generalize across the population. Lucas & Allen (2022) use intrinsic rewards as an alternative way to promote diverse behaviors. As an alternative to diversity-based approaches, Cui et al. (2021) propose to ground policies using hierarchies of agents, where a level-$k$ agent learns the best-response strategy to a level-$k-1$ agent. Similarly, Hu et al. (2021b) propose to iteratively optimize a policy against the optimal policy of the previous iteration. Through these hierarchies, the agents observe different levels of reasoning and can fall back to lower-level reasoning when playing with unknown partners. Both these works focus on the 2-player setting of Hanabi, and are unable to generalize to higher player settings. Finally, Hu & Sadigh (2023) propose a framework where the partner specifies what kind of strategy it expects the learning agent to play. Using pretrained large language models (LLMs), a prior policy is generated conditioned on this specification, such that the agent learns a strategy that is aligned with this policy. However, this prior policy is only conditioned on other agents' actions and heuristic instructions in contrast to our work where the language model operates on both observations and action avoiding the need to design hand-coded instructions. In all these works, agents need to learn how to play with others, on the same game-setting. In contrast, our R3D2 agent can play on multiple game-settings at once, and play with agents trained on other game-settings.

**RL for text-based games**   Recent successes in RL and natural language processing (NLP) have resulted in a surge of interest for developing RL agents based on text-based games. Examples include story-based games (Hausknecht et al., 2020), adventure games (Yin & May, 2019), and many others (Yuan et al., 2018; Murugesan et al., 2020; Wang et al., 2022). In such environments, the agent is presented with a textual description of a goal (Osborne et al., 2022). These interactive environments offer challenging and realistic training that requires a solid understanding of the language and the task. Moreover, connecting language with the physical world is critical to solving the task (Bisk et al., 2020; Bender & Koller, 2020). To address these, researchers have developed several RL-based agents that operate on text (He et al., 2015; Jain et al., 2019; Xu et al., 2020; Yuan et al., 2018). Language models have been used to propose action candidates (Jang et al., 2021; Yao et al., 2020; Singh et al., 2021; Sudhakar et al., 2023). While these environments allow for many different and diverse behaviors and tasks, they focus on single-agent learning. For multi-agent systems, Park et al. (2023) investigate generative agents interacting with each other and the world through text. However, the focus of this work is to observe believable individual and emergent social behaviors from these agents, no learning is involved.

**Transfer Learning and Generalization in RL**   Transfer learning and generalization in multi-agent reinforcement learning present unique challenges due to complex agent interactions. While single-agent approaches focus on task-invariant features (Taylor & Stone, 2009; Finn et al., 2017) and curriculum learning (Bengio et al., 2009; Narvekar et al., 2020), multi-agent systems face additional challenges in adapting to different interaction patterns (Carbonell & Veloso, 2006; Da Silva & Costa, 2019). Recent work has explored environment design, with Team et al. (2021) and Dennis et al. (2020) demonstrating zero-shot generalization through procedurally generated environments, and Samvelyan et al. (2023) extending this to multi-agent coordination. Our work diverges from these approaches by using language to reformulate environments for cross-configuration generalization, building on language-based transfer advances (Andreas et al., 2017; Lee et al., 2022).

# 3 PRELIMINARIES

## 3.1 THE GAME OF HANABI

In Hanabi, 2-to-5 players work cooperatively to arrange cards in ascending order (from 1 to 5) for each color (typically five colors: red, blue, green, white, and yellow). Each player is dealt a hand of cards, but the twist is that they cannot see their own cards, only their teammates can see them. Players take turns, and on each turn, a player must choose to perform one of three possible actions. Either it gives a clue, plays a card, or discards a card. If the player *gives a clue*, it can provide one piece of information to another player about their hand (e.g., "These two cards are blue"). Giving a clue costs a *hint token*, and there are 8 hint tokens available. Alternatively, the player can decide to *play a card*, hoping it can be added to the sequence of cards on the table. If the card is correct (i.e., it extends one of the color sequences in ascending order), it is successfully played. If the card is wrong, it is discarded, and the team loses one of its 3 *life tokens*. Finally, the player can *discard a card* from their hand to gain a hint token back. This action is necessary when hint tokens run out, but it comes with the risk of discarding a crucial card. Each correctly arranged card results in 1 point, for a maximum of 25 points (all 5 cards of all 5 colors have been arranged correctly). The game ends either when all cards are arranged, all life tokens have been used, or the deck is empty.

## 3.2 MULTI-AGENT REINFORCEMENT LEARNING

In Hanabi, each player is dealt a number of cards, which only the other players can see. As such, Hanabi is a *partially observable* game: not all information is available to the player to make a decision. Each turn, a player can decide to either play or discard a card, or to give a clue. Players need to make decisions alone, making this a *decentralized* game. By playing cards in a specific order, players gain points as a team. Hanabi is thus fully cooperative. In the MARL literature, such a setting is modeled as a Decentralized Partially-Observable Markov Decision Process (Dec-POMDP) (Bernstein et al., 2002; Nair et al., 2003), formally defined as a tuple $G = \{S, A, P, R, \Omega, O, N, \gamma\}$, with the set of states $S$, the set of actions $A$, the transition function $P$, the reward function $R$, the set of observations for each agent $\Omega$, the observation function $O$, the number of agents $N$, and $\gamma$ as the discount factor. The game is partially observable, with $o^i \sim O(o \mid i, s)$ as agent $i$ 's observation of the global state, sampled from the (stochastic) observation function $O$. The game is also fully cooperative, thus agents share the same reward $r = R(s, \boldsymbol{a})$, conditioned on the joint action $\boldsymbol{a} = [a^i]_{i=1}^{N}$ and the global state $s$. At each timestep $t$, all agents are at the state $s_t$. Each agent has an action-observation history (AOH) $\tau_t^i = \{o_0^i, a_0^i, r_0^i, \ldots, o_t^i\}$, and selects action $a_t^i$ using a stochastic policy of the form $\pi_\theta^i (a^i \mid \tau^i)$. The transition function $P (s' \mid s_t, \boldsymbol{a}_t)$, conditioned on the joint action and the global state, transitions to the next state $s_{t+1}$. Under the joint policy $\pi$, we call $V^\pi(s) = \mathbb{E} \sum_t [\gamma^t r_t \mid \pi, s_t = s]$ the *value*, i.e., the expected sum of discounted rewards (or return). The policy that maximizes the value is said to be the optimal policy $\pi^* = \max_\pi V^\pi$. Closely related to the value is the $Q$-value $Q^\pi(s, a) = \mathbb{E} \sum_t [\gamma^t r_t \mid \pi, s_t = s, a_t = a]$.

In single-agent RL, Deep Q-Networks (DQN) (Mnih et al., 2015) learns $Q_\theta$, an approximation of $Q^*$ with a neural network parametrized by $\theta$. $Q_\theta$ is learned by minimizing $(Q_\theta(s, a) - (r + \gamma \max_{a'} Q_{\theta'}(s', a')))^2$, where $Q_{\theta'}$ is a copy of $Q_\theta$ updated regularly to stabilize learning. To extend DQN to the partially observable setting, Hausknecht & Stone (2015) propose Deep Recurrent Q-Network (DRQN), which uses recurrent neural networks to estimate $Q$-values based on the AOH, i.e., $Q^\pi(\tau, a) = \mathbb{E} \sum_t [\gamma^t r_t \mid \pi, \tau_t = \tau, a_t = a]$. This, combined with several modern best practices on top of DQN, including double-DQN (Van Hasselt et al., 2016), a dueling network architecture (Wang et al., 2016), prioritized experience replay (Schaul et al., 2016), and a distributed training setup with parallel running environments (Horgan et al., 2018) result in Recurrent Replay Distributed Deep Q-Networks (R2D2) (Kapturowski et al., 2019). Our proposed algorithm, which we explain in section 4, uses R2D2 as its foundation. In MARL, a straightforward strategy is to represent each agent as an independent single-agent learner, e.g., using R2D2. An agent can then be trained through self-play (SP) with copies of itself (Tan, 1993; Tampuu et al., 2017). Our proposed algorithm, R3D2, also uses SP to learn cooperative strategies.

## 4 METHODOLOGY

We present Recurrent Replay Relevance Distributed DQN (R3D2), a generalist Hanabi agent that adapts to novel partners and game-settings. R3D2 observes and interacts with its environment through natural language, which is known to improve transfer of learned behavior to other tasks. Through its network architecture R3D2 handles dynamic observation- and action-spaces. Combined with a distributed training regimen, it allows R3D2 to jointly learn diverse strategies for 2-to-5 player games, resulting in a robust, adaptive artificial player.

### 4.1 HANABI AS A TEXT-BASED GAME

As our first contribution, we frame Hanabi as a text-based game, due to recent successes of using language as medium for transfer learning (Radford et al., 2019b; Brown et al., 2020).

In the original Hanabi environment (Bard et al., 2020), the observation is represented as a bitstring encoding, i.e., a concatenation of bits representing the observation. To encode the game state as text, we use a template, of which an example can be seen in Figure 1. The template starts by listing the number of life and clue tokens, followed by a listing of the arranged cards, other players' hands, and the knowledge of the player's own hand. It also includes the hints given to other players. While the bitstring encoding captures all the information available for optimal gameplay, it greatly changes depending on the setting (2-to-5-player games). In contrast, our text encoding appears intuitive, and requires minimal modifications switching between game settings. Thus, it allows for state-space generalization, i.e., learning policies across different settings becomes easier. For example, adding a new hand to the game results in arbitrary modifications in the original bitstring encoding. In contrast, the text encoding only appends relevant information to the observation.

Secondly, we modify the action-space. There are 3 types of actions: play a card, discard a card and give a clue. Each type has a fixed number of concrete actions. A player can play or discard any of the cards in their hand (e.g., "I play card 1"). Or, a player can give a clue concerning a specific color or rank to any other player (e.g., "I reveal `blue` to player 2"). In the original environment, actions are encoded as a one-hot encoding of all the possible action-combinations. Instead, we encode the action with a keyword corresponding to the action-type, followed by the type's parameter (e.g. `reveal blue 2`). Clues are the only type of actions affected by a change in the number of players. With this encoding, the agent can generalize the behavior for each action-type, and easily extend clue-actions to other player settings. This motivates generalizations to actions not seen during training.

We select this template because it includes all the crucial information contained in the vectorized observation. We also perform ablation studies on different components of the template to measure its impact on the agent's capability to predict optimal actions, which we show in Appendix A.9. Using this textual version, we design an agent that takes advantage of this representation to improve generalization across players and settings.

### 4.2 R3D2 AGENT: HANDLING DYNAMIC STATE AND ACTION SPACE

With the dynamic representation resulting from Hanabi's text-based encoding, we propose an agent architecture (shown in Figure 2) that incorporates a (pretrained) language model to encode observations and actions into observation- and action-embeddings. To do so, we take inspiration from Deep Reinforcement Relevance Network (DRRN) (He et al., 2015), which propose a neural architecture for deep RL designed to handle action-spaces characterized by natural language. In contrast to R2D2, which outputs a fixed vector of size $|A|$, DRRN encodes the action in a separate action-embedding network. The corresponding $Q$-value is then computed as the inner product of the state- and action-embeddings:

$$Q(s, a) = f(s)^\top g(a),$$

where $f(s)$ is the state embedding and $g(a)$ is the action embedding. In doing so, DRRN is able to encode an arbitrary number of actions. Ma et al. (2022) also showed recently that attention-based architectures that jointly process a featurized representation of observations and actions have a better inductive bias for learning intuitive policies. As a downside, $|A|$ separate forward passes need to be performed on $g$ to compute all $Q$-values of a specific state $s$.

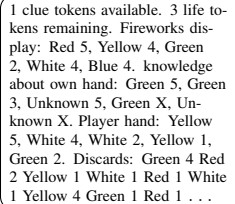

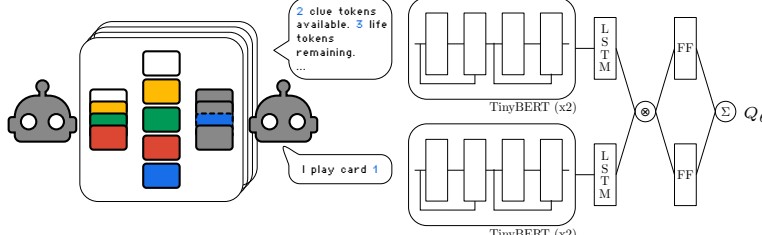

Figure 1: The template used for textual observations in Hanabi. It includes all necessary information to play the game, including life and clue tokens, visible hands, discarded cards, and hints.

Figure 2: An overview of the R3D2 architecture. R3D2 uses a separate head for observations and actions. Each head starts with 2 TinyBERT layers to encode the textual representation, followed by a LSTM layer to encode the previous timesteps. We use elementwise multiplication to combine both embeddings. This is then split into a separate value and advantage head, before being summed together to obtain $Q_\theta(\tau, a)$.

In our case, $Q_\theta$ depends on the AOH, resulting in a sequence of embeddings. This sequence is processed by a Long Short-Term Memory (LSTM) network, followed by a two-layer perceptron (MLP) to predict value and advantage estimates. These are then combined to estimate $Q$-values. We train our R3D2 agent via self-play, similar to IQL-based baselines in the literature (Hu & Foerster, 2019) to minimize the TD-loss. In section 5, we demonstrate that using an appropriate representation one can enable self-play to learn cooperative behavior with other partners, without relying on complex MARL methods.

The primary trainable components of R3D2 include the language model (LM), LSTM, and MLP. We utilize a two-layer TinyBERT (Jiao et al., 2020) as the LM due to its optimal balance between performance and inference time, which is crucial in RL settings with frequent interactions. Moreover, we performed preliminary experiments with different small LM to confirm this choice (see section 5.4), and to analyse the impact of pretrained weights, as well as the frequency at which to update the LM (see section A.2.1). It is worth noting that larger text encoders could also be integrated; however, the inference cost of large language models can become a bottleneck (Kaplan et al., 2020), which remains an open area of research.

We follow the same training procedure as R2D2, by using large number of parallel environments to gather trajectories and prioritized experience replay to sample transitions to update $Q_\theta$. Moreover, we designed R3D2 to support environments with varying numbers of players, ranging from 2 to 5 participants. To achieve this, multiple parallel actors take actions in these settings, while a shared replay buffer gathers the trajectories from all actors together. Although the agent itself is agnostic to the number of players, we adapt the replay buffer to handle sequences of different lengths. To address this, we pad the token trajectories with zeros, ensuring a consistent buffer structure across all trajectories, regardless of the number of players.

## 5 EXPERIMENTS

In this section, we analyse the generalization performance of R3D2 on Hanabi. Hanabi is a challenging task, which requires to learn conventions with other players to reach a high score. When changing the number of players participating in a game, the strategy to reach a high score changes, even though the rules of the game remain the same. For example, the number of allowed clues remains 8 in a 2-player and 5-player setting, despite having more players that require hints in the 5-player setting. Despite requiring to change strategies, some of the learned conventions remain the same, and should be transferable from one setting to another. For example, providing the clue "this card is a card of rank 5" tells the player that they absolutely cannot discard that card, as there is only one such a card for each color, and it is required to play it to reach a maximum score. We aim to learn how specialized the conventions of agents that have played together during training are, by evaluating them in a ZSC fashion with agents they have never encountered before. We also aim to evaluate just how much these conventions can be carried on from one setting to another, by pairing agents that have been trained on different settings together. Finally, since our R3D2 agent is flexible enough to play on any setting of the game, we also train our R3D2 agent on all game settings at

the same time. We assess the benefits of playing on more diverse types of gameplay, and how this improves the agent's cooperation abilities with others.

## 5.1 EXPERIMENTAL SETUP

We train single-setting R3D2 agents for each setting, i.e., from 2-player games to 5-player games. We call these agents R3D2-S (*single* setting). Analogously, we call R3D2-M (*multiple* settings) the R3D2 agent trained on all game-settings concurrently. For comparison, we train 3 different baselines on the original, vectorial version of the Hanabi environment. The first baseline is Independent Q-Learning (IQL) (Tan, 1993; Tampuu et al., 2017). IQL is still frequently used as a baseline for Hanabi, as it serves as the foundation of many state-of-the-art MARL Hanabi algorithms, including the other baselines we use in this work. We call this baseline R2D2 as it is based on R2D2 agents trained independently through self-play. It shows strong performance with its training partners, having learned highly specialized conventions through self-play. However, this also means that it typically does not play well with other partners, who do not necessarily follow the same conventions. For our second baseline, we use Other-Play (OP) (Hu et al., 2020). In Hanabi, playing a game with permuted colors of the cards does not change the game, i.e., there exist symmetries in the game that the policy should be invariant to. OP learns such policies by exploiting known symmetries of the Dec-POMDP. This avoids over-specialized conventions that would break on different, but symmetrical (and thus equivalent) states of the game. We call this R2D2-based OP agent: R2D2-OP. Our final baseline is Off-Belief Learning (OBL) (Hu et al., 2021b). OBL assumes past actions where taken by a fixed policy, different from its own. OBL then converges to an optimal grounded policy, which can in turn be used to ground a higher-level policy. In our experiments, we use OBL after 4 levels of grounding, as this is the highest level of grounding used in their original work. To highlight that this baseline is also R2D2 based, we call it R2D2-OBL. All these 3 baselines are value-based, learning a $Q_\theta$ network. To ensure a fair comparison with R3D2, all baselines use R2D2 as a basis, which contains several modern best practices for learning $Q_\theta$. Note that although we train and test R3D2 in settings ranging from 2 to 5 players, we utilized R2D2-OP and R2D2-OBL checkpoints from the original paper, which focused exclusively on the 2-player setting, as it was the only configuration examined in those studies. To isolate the contributions of our two key innovations - using language models for state representation and handling dynamic action spaces - we created an intermediate baseline called R2D2-text. This agent combines R2D2's fixed action space with R3D2's text-based state representation.

All baselines use the same neural network architecture, a recurrent neural network which takes a vectorial observation as input and outputs the $Q$-values for all possible actions. Details about the network architecture and hyperparameters are described in Appendix A.4. Each agent is trained for 2000 epochs with each epoch corresponding to 500 batch updates. For each experiment, we run three different seeds. For each evaluation setup, we define the performance of a team of agents as the average performance over 1000 games.

## 5.2 ZERO SHOT COORDINATION TO NOVEL SETTINGS

As a first set of experiments, to showcase our agent's ability to transfer policies across varying configurations, we evaluate its performance in different game settings. For a game with $n$-players, we select a subset $i < n$ of players that have been trained on $n$-player games, and we pair them with players that have been trained on $m$-player games (each player is a different seed). By aggregating all subsets $0 < i < n$, we can assess how well the agents of $m$-player games generalize to $n$-player games in a zero-shot manner. Each combination of $n$ and $m$ is shown in Figure 3. As a reference point, we showcase the self-play (SP) performance of R3D2 (leftmost bar on each plot). Finally, we also showcase the cross-play performance of $n$-player agents combined with R3D2-M (in blue). Finally, note that, when $n = m$, this results in standard intra-cross-play (e.g., `5p-XP` on the rightmost plot).

We make several observations. First, unsurprisingly, increasing the number of players results in lower overall performance, as strategies become more complex. Next, R3D2-M learns competitive strategies for all settings despite receiving the same training budget as single-setting algorithms, and is able to play well with single-setting players. This shows that knowledge from one setting is useful for other ones, and that the network architecture used by R3D2 allows for effective transfer of

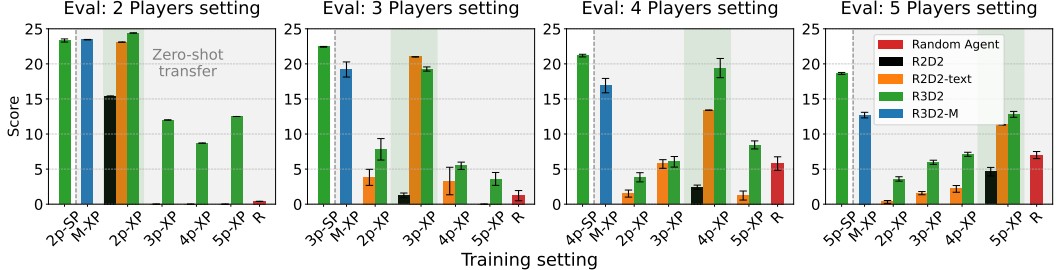

Figure 3: Policy Transfer - Zeroshot setting. Each subplot shows the evaluation setting for a $n$-player game. Each bar combines $0 < i < n$ agents trained on a different setting, with $n - i$ players trained on $n$-player games. R3D2 agents demonstrate strong zero-shot generalization to novel settings. Moreover, R2D2-text seems to be unable to match R3D2's transfer performance specially when transferring from a setting with large number of actions to smaller action space.

knowledge across settings. Additionally, R2D2-text is unable to match R3D2's transfer performance specially when transferring from a setting with large number of actions to smaller action space. Finally, R3D2's standard cross-play scores remain high, despite using self-play during training. The cross-play scores drop compared to self-play as the number of players increase, but that it because this results in more unique players playing together for the first time. in comparison, R2D2-text systematically has a lower cross-play score than R3D2, despite having the same action-space as during training. This shows that, while textual observations help for learning generalizable policies, incorporating dynamic action-spaces is inportant as well for same-setting scenarios. We refer to Appendix A.1 for additional comparisons on cross-play.

## 5.3 ZERO SHOT COORDINATION TO NOVEL PARTNERS

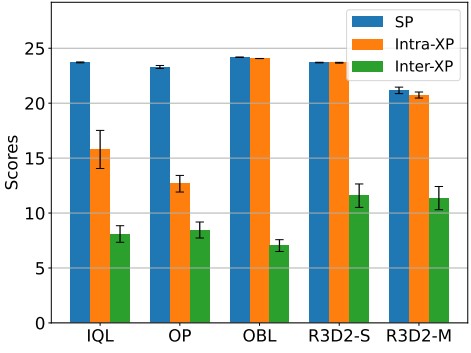
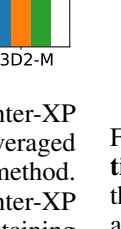
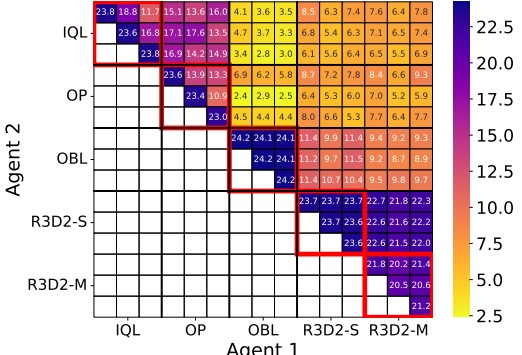

Figure 4: Selfplay, intra-XP and inter-XP performance in 2-player setting averaged across three independent seeds per method. R3D2 achieves significantly better inter-XP compared to the baselines while maintaining a competitive SP and intra-XP.

Figure 5: 2 player **zero-shot coordination** matrix between different methods and three independent seeds per method. R3D2 achieves better inter-XP with IQL, R2D2-OBL and R2D2-OP.

Now, we compare the robustness of R3D2's learned policies when partnered with novel agents, for the same game setting (i.e., cross-play) with other baselines. We not only compare the cooperation capabilities of different seeds of the same learning algorithm, but also of different algorithms with each other. We make 2-player teams composed of all combinations of seeds of all training algorithms, and evaluate their performance. The aggregated scores are shown in Figure 4. As a reference, we also show the performance when playing in self-play (SP) mode. The results of each individual combination of teammates results in the matrix of scores shown in figure 5. We note that the submatrices highlighted in red concern *intra-algorithmic* cross-play (intra-XP) (with their diagonal comparisons of the same seed, which is equivalent to self-play), while all other comparisons concern *inter-algorithmic* cross-play (inter-XP). We start with the observation that R2D2 and

R2D2-OP exhibit a lower intra-XP performance than self-play. While this is expected for R2D2, it is surprising for R2D2-OP, which is explicitly designed to cooperate well in the intra-XP setting. We believe this is because some of R2D2-OP's runs might not have exploited the environment symmetries well enough. The inconsistent behavior of R2D2-OP agents were also reported in Hu et al. (2020). Since this is the way R2D2-OP learns robust policies, failing to exploit these symmetries would results in a training procedure similar to R2D2. This would explain lower intra-XP scores, and would also explain why R2D2-OP cooperates surprisingly well with R2D2 in the inter-XP setting. On the other hand, R3D2's intra-XP performance is on par with self-play, even though it has been trained through self-play, similarly as R2D2. While R2D2's performance quickly degrades when paired with other seeds, this is not the case for R3D2, which confirms that self-play can lead to robust strategies, provided they use the appropriate representation. We believe that R2D2-OP, by exploiting known symmetries of the game, learns that different permutations of the vectorial version of Hanabi are equivalent. Trying to make sense of this vectorial representation thus contributes to R2D2-OP's cross-play abilities. In contrast, in our textual representation, symmetrical observations are the same but for a few tokens. Symmetries are thus implicitly tackled by R3D2.

The aggregated scores are shown in Figure 4. In general, while our agents achieve competitive performance in self-play and intra-XP, R3D2-based agents demonstrate superior inter-XP performance when paired with R2D2-OBL agents. This suggests R3D2 learns more robust and general strategies, in contrast to R2D2 and R2D2-OP which tend to learn brittle, specialized conventions. The strong performance with R2D2-OBL, which is known for learning human-compatible strategies (Hu et al., 2021b), indicates that R3D2 develops more natural and transferable coordination patterns. Interestingly, R3D2-M seems better at inter-XP than R3D2-S. Having been trained on multiple settings, R3D2-M has experienced more diverse strategies during training. We surmise that this diversity allows R3D2-M to be prepared to a wider range of novel policies, leading to higher overall collaboration. Next, we note that R2D2-OBL collaborates better with R3D2 than any other baseline. We thus ask ourselves if it is R2D2-OBL that is flexible enough to adapt to R3D2 or the opposite. We aim to answer this question by comparing their performance with R2D2, the least flexible policy. When R2D2-OBL is paired with R2D2, their score is lower than when R3D2-S is paired with R2D2. Thus, R3D2-S seems to have a more flexible policy than R2D2-OBL. An analogous analysis can be made for R2D2-OP, where R3D2-S paired with R2D2-OP achieves a higher score than R2D2-OBL with R2D2-OP. Results shown in Figure 5 clearly demonstrate that R3D2, even though trained using self-play, learns more robust policies than methods that explicitly aim to learn policies for ZSC.

## 5.4 LANGUAGE MODEL VARIANTS FOR HANABI

LMs have shown promising reasoning and planning capabilities Radford et al. (2019b); Brown et al. (2020). Having a completely text-based game, one might expect a LM should be able to play the Hanabi game successfully. Therefore, before testing our R3D2 agent, we evaluate several LMs on the Hanabi tasks with the text-based Hanabi environment, to understand the difficulty of playing the game of Hanabi for current LMs and establish a baseline for future improvements.

**Prompting and low-rank adaptation fine-tuning** Modern large LMs (LLMs) such as GPT-4 (OpenAI, 2023) and LLaMA-2 (Touvron et al., 2023) showcase remarkable zero-shot or few-shot generalization capacities, particularly in complex natural language tasks Kojima et al. (2022). First, we prompt GPT-4 to select actions for Hanabi, providing the textual observation and legal moves as context. Although it successfully avoids losing life tokens for extended periods, it struggles with optimal planning, limiting its ability to achieve scores higher than 3 or 4 [1]. Details about the various system prompts and game scores are available in Appendix A.6 and A.8. Next, we analyze how well a LLM fine-tuned on expert data would perform. For this, we fine-tune LLaMA-7B using low-rank adaptation (LoRA) (Hu et al., 2021a) on a dataset of expert data collected using an Off-Belief Learning agent (Hu et al., 2021b). Despite this tuning, the model performs poorly based on gameplay scores. Hu & Sadigh (2023) further supports the observation that current large LLMs are still far from independently solving Hanabi. For more detailed experimental results, we refer to Appendix A.8. These initial experiments seem to indicate that LLMs as-is are not sufficient to properly play Hanabi, and that learning to coordinate is required.

---

[1] We also tested the o3-mini model, which was released after the rebuttal, and observed significantly improved results. However, it still falls short of matching the performance of SOTA RL agents in gameplay. The code is available online as a notebook here).

**Full fine-tuning** Next we considered full fine-tuning of small LMs. Therefore, we focused on two types of language models—classifiers and generative models—serving as agents in the text-based Hanabi environment. We compare the impact of BERT-like architectures, i.e., BERT (Devlin et al., 2018a) and DistilBERT (Sanh et al., 2019), with the GPT-2 (Radford et al., 2019a) (classifier and generative) architecture.

To benchmark the performance, we select the optimal checkpoint for each LM based on gameplay scores. The best checkpoint is then subjected to 1200 runs in the Hanabi environment to handle variance and randomness, as depicted in Figure 6. Both the BERT and DistilBERT models demonstrate a commendable performance in the Hanabi gameplay, achieving a maximum score of 23 out of a possible 25. Their average gameplay scores hover around 10 during the gameplay. The GPT2-generative model has better top-k test accuracy. However, it fails short compared to the classification-based model in the overall gameplay score with $\sim 4.5$. We further tried using different percentages of training datasets to understand the role of data. Compared to 10% or less, when using 25% of the data, there is a sharp increase in the gameplay score. However, the performance plateaus for both 75% and 100% are indicative of reaching a saturation point. Also, we tried different BERT variants, and all are saturated to the same game score irrespective of the increase

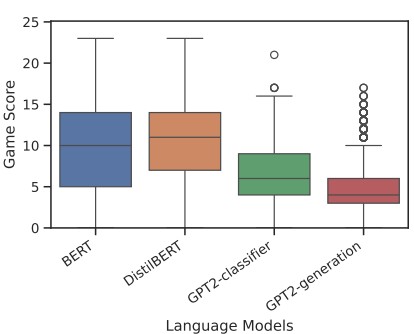

Figure 6: *The evaluation of language models performance as an agent is conducted via gameplay score, measured across various language models with 1200 game runs.*

in the parameter size. Finally, we also investigated the role of discarding information in the observation and found it didn't help much in the gameplay score. Refer to appendix A.9 for detailed ablation results on language models. Since the BERT architecture results in a higher performance than GPT-2, we use a BERT-like network in our R3D2 agent.

## 6 Conclusion and Future work

In this work, we show that learning through self-play can lead to robust policies, provided that the learning agent is trained with an adequate representation. We propose Recurrent Replay Relevance Distributed DQN (R3D2), that plays Hanabi with a textual representation of the game, and a player-agnostic neural network architecture. R3D2's intra-algorithmic cross-play score is on par with its self-play score, a first for Hanabi agents learning through self-play. Moreover, our experiments show that pairing R3D2 agents from different settings together can lead to collaborative success, with agents having been trained on more complicated settings being more capable in general than agents that have been trained on simpler settings, with less players in the game. Additionally, R3D2's player-agnostic architecture facilitates variable-player learning, enabling it to generalize strategies across various settings. This opens a new research avenue for exploring generalization across game settings, in addition to coordination with novel partners in MARL for complex cooperative games such as Hanabi.

Our approach, leveraging embedding and language models, is naturally adaptable to other text-based tasks. However, we acknowledge certain limitations - environments with continuous state/action spaces (like robotic control) or image-based inputs would require domain-specific adaptations. However, recent advances in language models are expanding the possibilities. For instance, Llama-2's specialized tokenizer demonstrates remarkable performance on numerical tasks by decomposing numbers into digit sequences (Touvron et al., 2023). An interesting direction for future work involves enhancing inter-setting cross-play evaluation, which we introduced and see as having significant potential. This area allows for further exploration of robustness by improving agents' adaptability across different game settings. Expanding the evaluation to include various combinations of replaced agents and different algorithms could yield deeper insights. Additionally, while Zero-Shot Coordination serves as a useful benchmark, it may lack realism. Exploring few-shot coordination (Nekoei et al., 2023) could be a promising research direction, where agents quickly adapt to new environments and partners, striving for consensus and effective collaboration in minimal episodes, offering a more dynamic approach to agent interaction in complex scenarios.

ACKNOWLEDGEMENTS

We thank the anonymous reviewers for their insightful comments and constructive suggestions that significantly improved the quality of this manuscript. The authors acknowledge the computational resources provided by Mila and the Digital Research Alliance of Canada. Hadi Nekoei is supported by NSERC and FRQNT Doctral scholarships. Sarath Chandar is supported by the Canada CIFAR AI Chairs program, the Canada Research Chair in Lifelong Machine Learning, and the NSERC Discovery Grant. This work was also supported by an IBM-Mila collaboration grant.

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

## A APPENDIX

### A.1 ZERO-SHOT COORDINATION TO NOVEL PARTNERS

To demonstrate R3D2's robustness, we report self-play and intra-XP performances of R3D2, IQL, and OP trained on 3-, 4-, 5-player game settings in Figure 7. IQL and OP achieve high self-play scores but perform poorly in cross-play. R3D2 variants, particularly R3D2-S, demonstrate more consistent performance across both metrics, maintaining scores above 15 points in all scenarios despite the general decline in performance as player count increases. R3D2-M This suggests that R3D2's training approach leads to more robust and adaptable agents, though at a slight cost to self-play performance compared to IQL and OP.

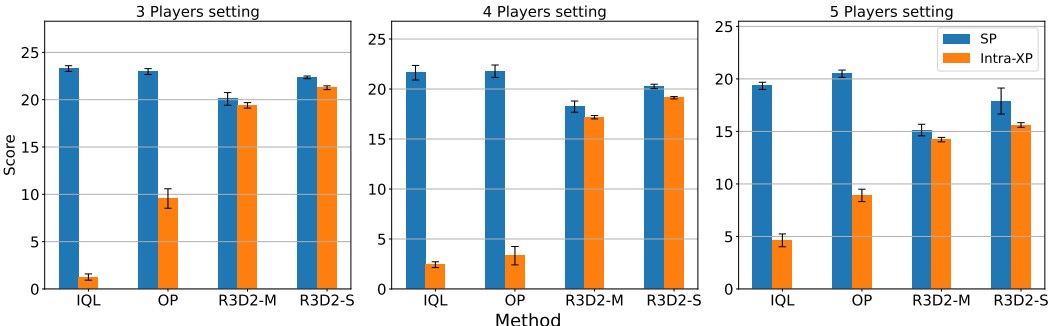

Figure 7: Performance comparison across different settings. The table shows both self-play (SP) and intra-cross-play (Intra-XP) scores for different methods in 3- , 4- and 5-player Hanabi settings. While IQL and OP achieve high SP scores but fail in Intra-XP, both R3D2 variants maintain consistent performance across both metrics, with R3D2-S showing particularly strong results.

### A.2 ABLATION STUDIES ON THE ROLE LANGUAGE MODELING

To better understand the impact of different components in R3D2, we conduct a series of ablation studies examining the role of language model pre-training, update frequency, and architectural choices. These experiments help isolate the contributions of our key innovations and validate our design decisions.

#### A.2.1 LM INITIALIZATION AND UPDATE FREQUENCY

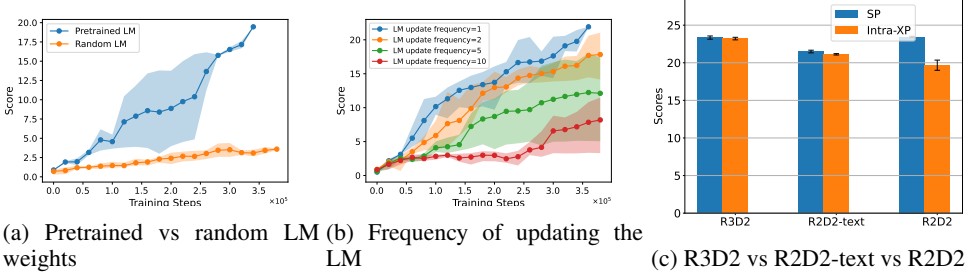

(a) Pretrained vs random LM weights  (b) Frequency of updating the LM  (c) R3D2 vs R2D2-text vs R2D2

Figure 8: Impact of pre-trained weights and update frequency on learning efficiency. (a) Performance difference between R3D2 agents trained with pre-trained language model (LM) weights versus randomly initialized LM weights, showing significant improvements in sample efficiency with pre-trained weights. (b) The effect of varying the frequency of LM updates, highlighting that frequent updates are critical for effective learning in the Hanabi environment.

We train two R3D2 agents in a 2-player Hanabi setting: one using a pre-trained language model (LM) and the other with the same architecture but randomly initialized LM weights. Figure 8a shows that learning from pre-trained weights significantly improves the sample efficiency. Additionally, we test updating the LM less frequently with periods of 1, 2, 5, and 10 training steps per LM update to examine whether the original pre-trained weights provide sufficient representations for playing

Hanabi or if fine-tuning is necessary. Our results, presented in Figure 8b, indicate that updating the LM parameters is essential for effective learning.

### A.2.2 Does the R3D2 performance comes from language model or the architecture?

As shown in Figure 8c, while R2D2-text achieves better intra-XP performance than the original R2D2, it still falls short of R3D2's capabilities. R3D2 matches R2D2's strong self-play performance while significantly outperforming both baselines in intra-XP scenarios. These results demonstrate that both innovations are crucial: text representation alone provides some benefits for generalization, but the combination with dynamic action space processing is necessary to achieve robust transfer to novel partners.

### A.3 R3D2 vs R3D2-M as the fixed partner

Building upon our previous analysis in Figure 3, where we demonstrated R3D2's zero-shot transfer capabilities with at least one specialized agent, we further investigate the generalization capabilities of our multi-task variant, R3D2-M. We conduct a comparative analysis by positioning R3D2-M as the fixed partner and evaluating its cross-play performance with partners trained across different game settings. Figure 9 presents the performance comparison between R3D2 and R3D2-M when paired with R3D2-S partners trained on various settings (indicated on the x-axis). The results reveal comparable performance patterns between the two variants, with R3D2 exhibiting superior performance in certain scenarios (e.g., 2-player setting) while R3D2-M demonstrates stronger capabilities in others (e.g., 5-player setting). This balanced performance profile suggests that R3D2-M maintains robust generalization capabilities, achieving performance levels comparable to its single-task counterpart, R3D2-S, despite being trained on multiple settings simultaneously.

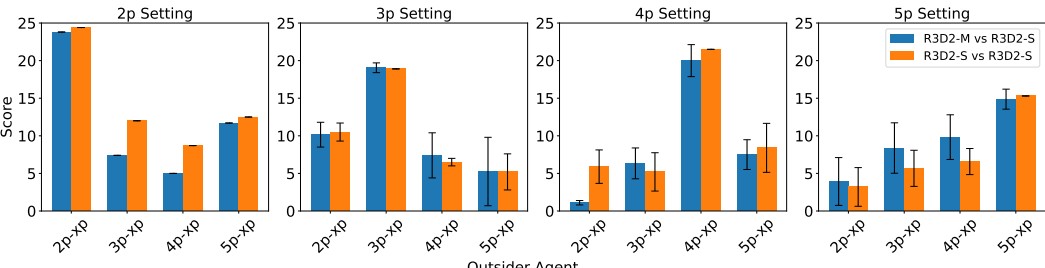

Figure 9: Policy Transfer - Zeroshot setting. Each subplot shows the evaluation setting for a $n$-player game. Each bar combines $0 < i < n$ agents trained on a different setting, with $n - i$ players trained on $n$-player games. Comparing R3D2 and R3D2-M's cross-play performance with R3D2-S partners trained on different settings (Figure 9), we observe complementary strengths: R3D2 excels in 2-player settings while R3D2-M performs better in 5-player scenarios. Despite being trained on multiple settings simultaneously, R3D2-M achieves comparable performance to its single-task counterpart, demonstrating robust generalization capabilities.

### A.4 R3D2 training setup

Here we provide all the experiment details and hyper-parameteres used to train R3D2 agents as shown in Table 1. To train R3D2 agents, we did not run a hyper-parameter tuning and relied on the HPs used by Hu et al. (2021b).

### A.5 Software details

The code was implemented using PyTorch, and pre-trained language models were loaded using Huggingface. To gain insights for this paper, we employed Weights & Biases (Biewald, 2020) for experiment tracking and visualizations. Lastly, plots are created using the seaborn package. For RL algorithms, we used OBL agent (Hu et al., 2021c) to collect the expert trajectory and forked official instruct-rl codebase[2] to train the algorithm.

---

[2]https://github.com/hengyuan-hu/instruct-rl/tree/main

Table 1: Hyper-Parameters for R3D2 agents.

| Hyper-parameters | Value |
|---|---|
| `# replay buffer related` | |
| `burn_in_frames` | 10,000 |
| `replay_buffer_size` | 50,000 |
| `priority_exponent` | 0.9 |
| `priority_weight` | 0.6 |
| `max_trajectory_length` | 80 |
| `# optimization` | |
| `optimizer` | Adam |
| `lr` | 6.25e-05 |
| `eps` | 1.5e-05 |
| `grad_clip` | 5 |
| `batchsize` | 64 |
| `# Q learning` | |
| `n_step` | 1 (R3D2) |
| `discount_factor` | 0.999 |
| `target_network_sync_interval` | 2500 |
| `exploration` $\epsilon$ | $\epsilon_0 \ldots \epsilon_n$, where $\epsilon_i = 0.1^{1+7i/(n-1)}, n = 80$ |

## A.6 PROMPTING DETAILS

**System prompt 1:** *"You are an expert Hanabi player"*

**System prompt 2:** *"You are an expert Hanabi player focused on maximizing team coordination and achieving high scores with minimal mistakes. Follow these principles: Efficient Clue-Giving: Provide clues that give maximum information, using finesse and double clues to benefit multiple players. Deduction: Track played/discarded cards and deduce your own cards based on clues and game state. Avoid discarding critical cards. Disciplined Play: Play and discard safely, minimizing risk while optimizing the team's progress. Team Coordination: Follow team conventions and use subtle cues (timing, actions) to communicate intent without verbal clues. Score Maximization: Manage clue tokens and pace the game to ensure enough clues for critical moments. Avoid getting bombed!"*

## A.7 DATASET DETAILS

The dataset is acquired through self-play mode, utilizing a pre-trained OBL agent in the Hanabi game. Trajectories are filtered selectively with a gameplay score exceeding 20. Then, these trajectories are broken down into state-action pairs to suit language model training. During the initial data exploration, we found the action categories are imbalanced as shown in 10, hence the language model overfits to discard 4 based on the confusion matrix for the prediction. To avoid that, we did categorical sampling consisting of 2200 samples per action type, aggregating to 44,000 instances. Then we checked for duplicate states and dropped them, there were approximately 100 duplicates as this could mislead the model's learning. After which, 10% of the dataset is reserved for testing by random sampling. Further, the dataset is split into 90% for train and 10% for validation.

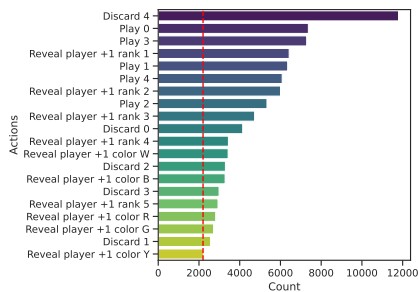

Figure 10: *Visualizing the number of actions available in the dataset to create a diverse dataset of Hanabi gameplay in the form of text.*

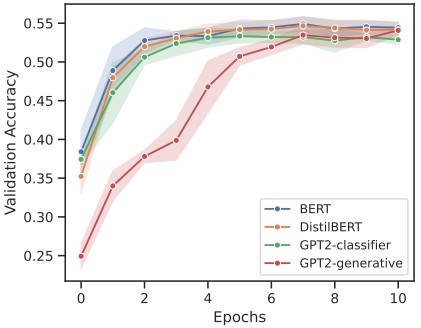 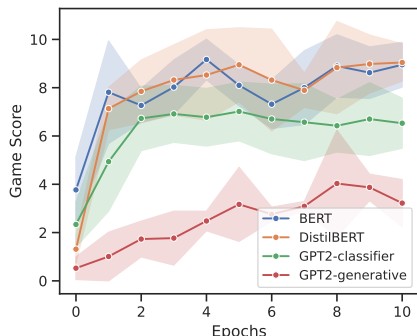

Figure 11: Learning graph for (a) Validation accuracy plotted against(b) Game play score, for each epoch for different language model providing insights into the observed trends during the training process.

Table 2: GPT-4 and o3-mini performance with different system prompts on text-based Hanabi averaged over 3 seeds. * including the final scores before getting bombed.

| Method | System Prompt 1 | System Prompt 2 |
|---|---|---|
| GPT-4 | $0.0 \pm 0.0$ | $0.0 \pm 0.0$ |
| o3-mini | $7.66 \pm 3.13$ | $12.0 \pm 0.47$ |
| GPT-4*(% Bombed) | $3.0 \pm 0.0$ (100%) | $2.34 \pm 0.27$(100%) |
| o3-mini*(% Bombed) | $9.0 \pm 2.05$(33.3%) | $12.0 \pm 0.47$(0%) |

### A.7.1 LANGUAGE MODEL SETUP

The model's finetuning process begins with a set of train-
ing instances, denoted as $(S, A)$ drawn from the dataset $\mathbb{D}$ where $S \in \{s_0, s_1, .., s_n\}$ and $A \in \{a_0, a_1, .., a_n\}$. Within this set, $s$ and $a$ represent a state and its corresponding noisy labelled action, respectively, and $n$ represents the number of examples in the dataset. The training objective of BERT, DistilBERT, GPT2-Classifier is,

$$L_{CCE} = -\frac{1}{N} \sum_{i=1}^{N} \sum_{j=1}^{C} a_{ij} \log(\hat{a}_{ij}) \qquad (1)$$

Where $N$ is the batch size. $C$ is the number of classes. $a_{ij}$ is the true probability of class j for the i-th example in the batch and $\hat{a}_{ij}$ is the predicted probability of class j for the i-th example in the batch.

The training objective of GPT-2 Generative is to minimize the cross-entropy loss, denoted as $\mathcal{L}$, and do the finetuning of the model. The cross-entropy loss is mathematically defined as follows:

$$\mathcal{L}_{LLM} = -\mathbb{E}_{(S,A) \sim D} \log p(A|S) \qquad (2)$$

Where $p(S|A)$ represents the conditional probability of predicting an action $A$, given the state $S$. The goal is to optimize these parameters, by minimizing the cross-entropy loss. We finetune the model to generate responses that better align with Hanabi game. The learning graph of validation accuracy with the game play score for each epoch is logged to understand the trend in the Figure 11(a,b). Mostly the Validation score and game score is getting saturated at around 4th epoch.

### A.8 HOW GOOD LLMS ARE IN PLAYING HANABI?

First, we evaluate GPT-4's ability to play Hanabi using our text-based format. While GPT-4 demonstrates basic game understanding by avoiding catastrophic moves, it achieves only rudimentary

scores of 3 points out of 25 (2), highlighting the limitations of pure language models in strategic planning.

To adapt the LLaMA to the gameplay, we use Low-Rank Adaptation, or LoRA (Hu et al., 2021a), which learns a low-rank decomposition matrices into each layer of the transformer architecture and freezes the pre-trained model weights. Thereby, significantly reducing the trainable parameters. We conducted fine-tuning experiments with LLaMA-7B weights with classifier using varying data sizes [200, 500, 1000] and LoRA ranks [32, 64, 128] for 10 epoch. Despite these parameter variations, the gameplay scores remained suboptimal level of around one as shown in 12. This highlights the challenges in achieving effective gameplay performance for current large langue model on playing hanabi.

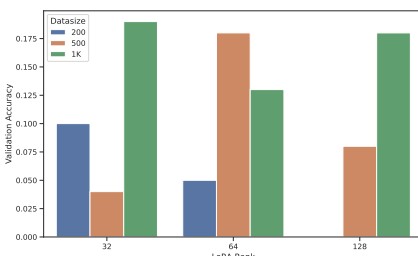 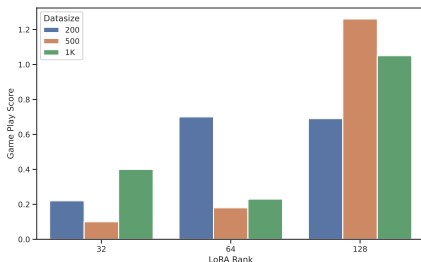

Figure 12: *Evaluation of Low-Rank Adaptation (LoRA) in LLaMA-7B finetuning, showcasing the impact on a) Validation Accuracy and b) Game Play Score. The experiments involve varying data sizes [200, 500, 1000] and LoRA ranks [32, 64, 128].*

## A.9 ABLATION STUDIES

### A.9.1 THE ROLE OF SCALING THE DATASET AND DIFFERENT MODEL VARIANTS

The dataset size emerges as a pivotal factor influencing gameplay scores. As the amount of training data increases there is a gradual increase in validation and the gameplay score. When the training percentage is equal to or less than 10% the games scores were poor ranging around 1 out of 25. In contrast, the gameplay score sharply increases when using 25% of the data as shown in 13b. Nevertheless, the performance plateaus at a game play score of approximately 9 for both 75% and 100%, indicative of reaching a saturation point, affirming the sufficiency of the dataset size for effective model training.

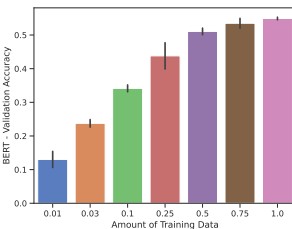 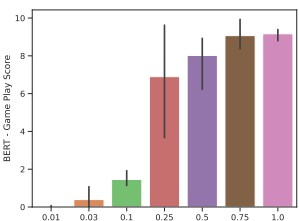 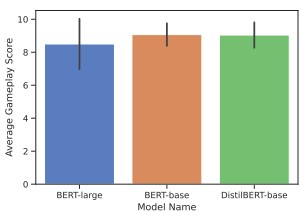

Figure 13: *Analysis of the impact of training data amount on BERT, examining a) BERT Validation Accuracy, b) BERT Game Play Score across different percentages of training data, and c) BERT model variants with varying parameter sizes.*

In our experimentation, we varied the model parameter sizes—ranging from DistilBERT with $66M$ parameters to BERT-base-uncased with $110M$ parameters and BERT-large-uncased with $340M$ parameters. We observed that DistilBERT achieves a competitive gameplay score of approximately 8.7 after 600 game runs 13c. On top of the performance considering the fast inference and low memory usage, DistilBERT was chosen as a candidate for integration with reinforcement learning through distillation.

### A.9.2 THE ROLE OF DISCARD INFORMATION

We examined the impact of incorporating the discard pile into the observation. Surprisingly, we discovered that utilizing the discard pile did not contribute to any improvement in game scores as show in the Figure 14. Rather, it resulted in a doubling of the sequence length of the language model. Given the need for fast inference in the reinforcement learning pipeline, we opted to exclude discard pile information from the observation during both language model training and inference. Nonetheless, there is a potential for heuristic-based approaches, to explore the idea of creating derived information from from the discard pile, potentially leading to a more concise sequence length and better game score.

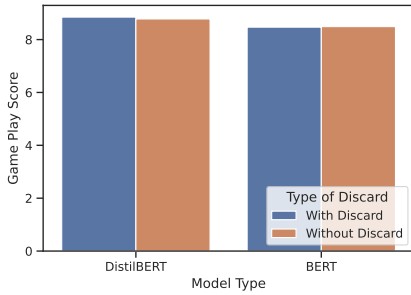

Figure 14: *Evaluation of the discard pile's role in the game is assessed by comparing game scores with the presence and absence of the discard pile in the observation during training.*

