# OpenReview forum: "A Generalist Hanabi Agent"
_ICLR.cc/2025/Conference — ICLR 2025 Poster_

### Official Review · Reviewer_u5D6 · 2024-10-30

**Soundness:** 2
**Presentation:** 3
**Contribution:** 2
**Rating:** 8
**Confidence:** 4

**Summary:**

This paper learns a generalist agent that is able to play any-card variants of Hanabi by casting the game to a text based game and utilize model architectures from language modeling to learn such policy with Q-learning

**Strengths:**

1. The idea of casting Hanabi as a text game for the purpose of learning a generalist agent is interesting. Language as a common interface for variants of games, or completely different games, is interesting, especially when the game primarily requires reasoning, which is the case in Hanabi.

2. If I understand correctly, the R3D2 agent is able to learn consistent policies, i.e. achieving high intra-AXP score and small gap between intra-AXP and self-play, without any additional assumptions that prior methods use. This is a cool result and probably the benefit of using language as observation and action? Although I am not sure exactly why this model is able to do that. See my question below.

**Weaknesses:**

1. Although this paper does a good job showing that we can train a single generalist agent for any-player Hanabi, which is an interesting contribution, one downside is that the paper is not able to have convincing results of the benefit of having a “generalist” agent, i.e. the benefit of going from R3D2-S to R3D2-M. Apart from the fact that the generalist agent does better in Figure 6, which it should because the single-variant agents are not trained on other variants of Hanabi, I only found this single comment on the benefit of R3D2-M:
> Interestingly, R3D2-M learns decent strategies for all settings, despite receiving the same training budget as single-setting algorithms. This shows that knowledge from one setting is useful for other ones, and that the textual representation allows for effective transfer of kenowledge across settings.

However, I don’t find this to be convincing. If the generalist agent is given the same amount of data as the sum of the 4 single-variant agents, will it outperform the single agents? I think this is crucial for making such a claim and I do believe that would make this paper stronger to the general audience beyond MARL.

\
2. The high inter-AXP results are not super exciting because:
* IQL is known to produce strange conventions that is hard to coordinate with;
* OP has significantly higher inter-AXP score with IQL than any other methods, indicating that the reproduced OP is probably as brittle as the IQL agents, which is totally possible as some OP implementation uses tricks such as hiding the last action from the observation to regularize more;
* OBL coordinates badly with the OP agents, indicating that they are very different;

With the arguments above, a high inter-AXP score here would just translate to better coordination with  IQL/OP than OBL, which is a useful result but not necessarily an important result because eventually we want policies that coordinates better with human-like policies (OBL) rather than inhuman policies (IQL). Having a human study as in [1], a paper in a comparable conference (ICML) on Hanabi that this paper cited, would greatly address this.

[1] Language Instructed RL for Human-AI Coordination.

**Questions:**

1.
What are the agents on the x-axis of Figure 6? Are they R3D2-k agents trained for each game variant? If so, why is the diagonal not the same as R3D2-SP? The notation and explanation around this figure could be improved. For example, you are using XP to refer to cross-play everywhere else, but use CP here, which could be a bit confusing.

2.
> In contrast, in our textual representation, symmetrical observations are the same but for a few tokens. Symmetries are thus implicitly tackled by R3D2.

Why is this the case? For example, considering two hands R1, R2, B5, B3, G4 and B1, B2, R5, R3, Y4 (R, G, B, Y for different colors), these two hands are symmetrical up to a color permutation used in OP, but they are more than a few tokens apart in the language description. The color permutation will also apply on other parts of the observations such as the firework pile so I would expect the language observation to be quite different? Am I missing something? If I did not miss something, I would be curious why the policy does not learn to use arbitrary color coding as a play strategy.

---

> ### Author Response · Authors · 2024-11-20
>
> Thank you for your thoughtful review and detailed feedback. We are pleased that you found the idea of casting Hanabi as a text-based game and using language-based representations intriguing, especially as a means to reason across diverse game settings. We also appreciate your recognition of R3D2's ability to achieve consistent policies and your interest in the potential of our generalist agent approach.
> You have raised some important concerns and questions, which we address below.
>
> > The paper is not able to have convincing results of the benefit of having a “generalist” agent, i.e. the benefit of going from R3D2-S to R3D2-M.
> If the generalist agent is given the same amount of data as the sum of the 4 single-variant agents, will it outperform the single agents?
>
> We would like to remark that a major benefit of R3D2 is that it is the first one to even be able to play on different game-settings. OBL, OP and IQL all need to be retrained to accommodate different player-settings. R3D2-M allows us to play any game-setting competitively with the same agent, something that is simply not possible with the baselines. And, while R3D2-S can play on other game-settings thanks to its architecture and textual representation, its zero-shot performance is worse than R3D2-M.
>
> Despite this, we have resumed the training of both R3D2-S and R3D2-M which, due to computational constraints, was trained for less iterations than the baselines. We expand upon this in the general response **[About R3D2’s performance]**, but the gist is that, for 2-player settings, the performance of R3D2-S increases for both self-play (22.23 → 23.34) and intra-XP scores (19.66 → 23.23). This trend is also observed for R3D2-M, with 20.46 → 21.40 for self-play and 20.26 → 21.27 for intra-XP.
>
>
> > Symmetries are thus implicitly tackled by R3D2. Why is this the case?
>
> R3D2 leverages a language-based representation, which implicitly models symmetries and is inherently compositional (Colas et al., "Language as a Cognitive Tool to Imagine Goals in Curiosity-Driven Exploration"). This allows R3D2 to learn generalizable strategies by marginalizing over symmetries, effectively covering many equivalent conventions.
>
> > we want policies that coordinates better with human-like policies (OBL) rather than inhuman policies (IQL)
> > The high inter-AXP results are not super exciting
>
> Thank you for raising this point. R3D2 focuses on coordinating with novel partners and game settings using only self-play. We not only have high inter-XP across agents but also, **R3D2 achieves higher inter-XP scores with OBL agents while only having been trained with self-play**, suggesting that it learns **more human-like strategies**, as OBL has been shown to coordinate well with humans. Benchmarking with human evaluations is an important direction, and we leave that as future work. Additionally, we want to mention our method is orthogonal to OBL agents. In future work, it would also be interesting to combine R3D2 and OBL.
>
> > What are the agents on the x-axis of Figure 6? Are they R3D2-k agents trained for each game variant? If so, why is the diagonal not the same as R3D2-SP?
>
> Thank you for pointing this out! The numbers in the bottom part of the heatmap represent cross-play scores, and the x-axis corresponds to an outsider agent introduced to the game. To make it even easier, we made a new plot in the revised paper (Figure 4). We have now divided the plot based on the player setting, where each sub-plot represents a specific setting. The **grey area** highlights the cross-play results.
>
>
> **In summary, we have addressed your concerns by providing additional training iterations for R3D2, improving our explanations regarding the benefits of R3D2-M, and reformulating parts of the manuscript for clarity. We kindly ask you to reconsider your rating in light of these improvements and the novelty of our contributions. We believe our work opens exciting directions for future research and contributes meaningfully to the MARL community. If we have not addressed your concerns to satisfaction, we would be glad to provide additional details.**

---

> > ### Comment · Reviewer_u5D6 · 2024-11-25
> >
> > Thank you for taking the effort to improve this paper. The newly added R2D2-text is helpful.
> >
> > I am very confused by Figure 4.
> > * What exactly is the meaning of the x-axis? You mention "We replace one to n−1 agents with outsider agents
> > shown on the x-axis", but the x-axis is 2p-xp, 3p-xp ....
> > * For example, for the the green bar in the 3p-xp column of  2 Players setting, are you pairing  R3D2-3-player with R3D2-2-player?
> > * In 3/4/5 player setting, why is the 3p-sp so different from the 3p-xp (both green bar)? I thought they are the same.
> > * What about the performance of xp(R3D2-M, R3D2-3-player) in a 2 player setting?
> >
> > Also I don't think the explanation around line 438 makes sense anymore for the new plot.
> > > As illustrated in Figure 4, for each game setting on the x-axis (e.g., a 3-player game), we replace one agent with an outsider agent, indicated on the y-axis.
> >
> > ---
> >
> > If the focus of the paper is on training a generalist, I think it misses some baselines/ablations on the generalist side of research. For example:
> > * Baseline: A non-text based agent that operates on the observation & action space that is the superset of all 5-variant and simply mask out invalid parts for different variants. Or tokenize the input space (a card becomes a token etc...), and use a transformer to represent the policy.
> > * Benefit of generalist in transfer: Is it possible to train the R3D2-M on 3 out of the 4 setting and play it in the remaining setting zero-shot?

---

> > > ### Author Response · Authors · 2024-11-28
> > >
> > > > I am very confused by Figure 4.
> > >
> > > We rephrased the explanation about Figure 4 in the paper, and corrected the outdated parts in the manuscript (thank you for pointing this out to us!).
> > >
> > > Figure 4 summarizes cross-play across different game-settings. Each subplot shows one specific game-setting (e.g., 5-player game setting). For this setting, we want to analyse how agents that have been trained on a different setting would be able to cooperate with agents from that game setting (e.g., if we combine 3-player agents with 5-player agents in a 5-player game, how do they cooperate with each other?). Each bar in the gray area from each subplot is one such combination. To address is how many of these 3-players to select in the 5-player game, we simply aggregate across all possible variants (1 3-player agent with 4 5-player agents, 2 3-player agents with 3 5-player agents, …, 4 3-player agents with 1 5-player agent). This is what we meant by *one to n-1 agents*, we apologize for the confusion. Note that each player is a different seed, so these agents are playing zero-shot with each other (all bars with XP on the x-axis represent cross-play agents). We added a shaded green area where it shows standard intra-XP scores commonly reported in the literature.
> > >
> > > >  why is the 3p-sp so different from the 3p-xp (both green bar)? I thought they are the same.
> > >
> > > SP is provided as reference, and points to the agents playing in self-play: the same-seed agent plays with itself. Since it has been trained through self-play, it has learned specialized strategies. By playing with itself, it can fully take advantage of these strategies. But, when paired with other seeds (XP), these strategies do not function as well anymore. The more players there are in a game, the more different strategies are mingled together. However, what we see is that, **despite being trained in self-play**, R3D2 agents can cooperate with others. This is in stark contrast to typical self-play methods, which do not generalize well. Using textual observations is one big reason for this generalization (shown by R2D2-text), but it is also due to the dynamic action-space, especially for generalization across game-settings. **For completeness, we have also evaluated cross-play settings of other baselines on 3,4,5P settings (see Appendix A.6).**
> > >
> > > > What about the performance of xp(R3D2-M, R3D2-3-player) in a 2 player setting?
> > >
> > > This is indeed not present in the plot. In Figure 4, we always assume there will be at least one specialized player in the game (a R3D2-2-player in a 2-player setting). So we compare R3D2-M (in cross-play) with specialized players only. This is an interesting suggestion, though. We ran these evaluations by replacing specialized R3D2-S agents with R3D2-M. Here is a summary of the results. (Refer to Appendix A.8 for a more complete figure including both R3D2-M and R3D2-S). The results reveal comparable performance patterns between the two variants suggesting that R3D2-M maintains robust generalization capabilities, achieving performance levels comparable to its single-task counterpart, R3D2-S, despite being trained on multiple settings simultaneously.
> > >
> > > | Setting | 2 Players (R3D2-M) | 3 Players (R3D2-M) | 4 Players (R3D2-M) | 5 Players (R3D2-M) |
> > > |---------|-----------|------------|------------|------------|
> > > | R3D2-2 | 23.80 | 7.40 | 5.00 | 11.70 |
> > > | R3D2-3 | 10.15 | 19.05 | 7.40 | 5.25 |
> > > | R3D2-4 | 1.10 | 6.33 | 20.00 | 7.50 |
> > > | R3D2-5 | 3.93 | 8.38 | 9.83 | 14.88 |
> > >
> > >
> > >
> > > > Baseline: A non-text based agent that operates on the observation & action space that is the superset of all 5-variant and simply mask out invalid parts for different variants. Or tokenize the input space (a card becomes a token etc...), and use a transformer to represent the policy.
> > >
> > > We thank you for these suggestions. Concerning the non-text agents, our experiments show that self-play agents do not play well in cross-play, even on same-setting games, with the same state-space as during training. Therefore, we do not expect a vectorized self-play agent to perform well on multi-game settings. Concerning the transformer, this is an interesting idea. We focus on textual observations, since it is known to help for transfer, but we definitely agree there might be some alternative representations that hold similar properties. We believe this to be out of the scope of our work, since we compare text versus non-text, but this might be an interesting avenue for extensions on our work.

---

> > > > ### Author Response · Authors · 2024-11-28
> > > >
> > > > > Benefit of generalist in transfer: Is it possible to train the R3D2-M on 3 out of the 4 setting and play it in the remaining setting zero-shot?
> > > >
> > > > Thank you for mentioning this. These variants are interesting and we are currently training them, but we believe that these do not diminish the fact that we are the first to propose an agent that can play on all Hanabi settings at the same time. Note that each experiment takes multiple days (using multiple GPUs), and training all combinations (training on 2&3P, 2&3&4P, 3$4P, …) is a time consuming process, so we will include these in the CR version, when they are ready.

---

> ### Comment · Reviewer_u5D6 · 2024-11-28
>
> Thank you for the detailed explanation. Now I understand the results much better, despite quite some effort. I would encourage the authors to improve the presentation to emphasize the benefits.
>
> I like that R3D2-3,4,5p models have a non-trivial performance in the 2-player setting despite being train on a single game setting.
>
> I am a bit surprised that in n-player setting, xp(R3D2-M, R3D2-k) where k != n  is not noticeably better than xp(R3D2-k, R3D2-k)  because I would expect the R3D2-M to be more "in-distribution"?
>
> I also like the new results on the GPT's performance on this task, although it is from other reviewer's comment.
>
> For this question below, I think one experiment, such as training on 2,4,5 but play on 3 player-setting, is sufficient.
> > Is it possible to train the R3D2-M on 3 out of the 4 setting and play it in the remaining setting zero-shot?
>
> I would like to increase my score to 8 to support the acceptance of this paper. Although I am more leaning towards somewhere between 6 and 8 but this paper has gotten so many reviews and I truly appreciate the authors effort to address them. So I pick the more encouraging side.

---

### Official Review · Reviewer_k3Nj · 2024-11-04

**Soundness:** 3
**Presentation:** 4
**Contribution:** 3
**Rating:** 8
**Confidence:** 5

**Summary:**

The paper aims to train a generalist Hanabi agent that can:
1) Play variants of Hinabi with different number of co-players
2) Play with novel co-players

To this end the authors contributions are:
1) A text based version of hinabi environment - which makes learning across multuple variants easier
2) A novel-ish algorithm R3D2, which is a recurrent network with a value function generated from a text embedding.

**Strengths:**

The problem is original - multi-task / multi-agent problems have not been tried since the XL land paper or Maestro (both citations which are missing in this work). Although this does truly feel like the limit of how much as a community we can care about hinabi.

I’m a massive fan of the evaluation protocol in this paper - its really nice to see cooperative marl analysis on cross-play and the introduction of intra-xp and inter-xp is a huge improvement.

The paper is well written and to a high quality.

**Weaknesses:**

The choice of baselines is really confusing to me, and makes it hard to understand the authors contribution. All baselines presented are trained and evaluated in the existing hinabi environment (the non-text based encoding). This makes it hard to disentangle the importance of the R2D2 style training, the RRN modification or the language based component underneath.  To my understanding, the method isn’t really novel - this is just R2D2 but they used the text embedding for both goals and actions - which is just a trick from the Continuous Control Literature and augmented the Recurrent network to be a recurrent relevant network. Also using a text based version of Hinabi is obvious in 2024, so i don’t see this as a significant contribution. Ablations would be appreciated to understand where the juice is actually coming from. I’d also like to just know if OBL agents in the language based environment generalize better.

The results lack qualitative analysis and as such, its really hard to understand whats happening here or the significance of the work, Is the hope that the method has learnt via a larger set of environments / tasks to have more robust policies.  If so what do these policies look like? Do they share underlying similar structures or methods. Could we also evaluate methods on OOD tasks (e.g 7 players hinabi or something not in the train?).
Its hard to interpret the results in Figure 6 - yes it seems R3D2 is robust to co-players changing but because i don’t have an equivalent evaluation with a team of IQL agents or OBL agents its very hard to evaluate if this is comparatively better.

The paper would be vastly improved if consistent LLMs were used throughout the section. LLMs have quirks, different training data and different methods. It feels hard to compare results in Section 5 with any proposed in Section 6 because the underlying models are very different. I also think that the experiments in Section 5 tell us more about those models (which are rather small number of parameters) and the difficulty of supervised learning here. The conclusions drawn are overreaching.

**Questions:**

1) “it successfully avoids getting bombed for extended periods, “ <- is this a common term in hinabi?

2) The Appendix is missing.

3) The Inter-XP metric is calculated with which other co-player? Figure 4.

4) Could analysis be provided of the compute costs / sample efficiency and limitations of each method? Its really hard to work out why this method is working at all, is this just better domain randomisation, is there some particular underlying set of hinabi skills that generalise well over n players.

---

> ### Author Response · Authors · 2024-11-20
>
> Thank you for your review and encouraging feedback. We are thrilled that you found the problem we tackle original and appreciated the novelty of our evaluation protocol, which we believe provides representative insights about the zero-shot capabilities of agents. Your recognition of the high quality and clarity of our paper is incredibly motivating, and we are grateful for your thoughtful insights.
>
> You have made some valuable comments, which we address below.
>
> >  XL land paper or Maestro (both citations which are missing in this work).
>
> Thank you for these references. We have expanded the related work (Section 2, line number - 155 to 159) to discuss these works.
>
> > hard to disentangle the importance of the R2D2 style training, the RRN modification or the language based component underneath. Ablations would be appreciated to understand where the juice is actually coming from.
> > Its really hard to work out why this method is working at all, is this just better domain randomisation, is there some particular underlying set of hinabi skills that generalise well over n players.
>
> This is a very valid point. We have **added additional ablation experiments (Section 6.2 and A.7)** to assess the role of language in R3D2. We kindly refer to our general comment on the  **[Role of language and language models]**. In summary, while a text representation provides some benefits for generalization, handling dynamic action spaces is equally crucial for successful transfer to novel partners.
>
> > I’d also like to just know if OBL agents in the language based environment generalize better.
>
> This is a really interesting proposition. The goal  of our work was to design a **simple, scalable, and robust** agent that requires no architectural changes across different player settings. Notably, **R3D2 achieves higher inter-XP scores with OBL agents while using only self-play**. However, our contributions can be seen as being orthogonal to OBL agents. We believe combining R3D2 and OBL would be a really interesting avenue for future work. Please refer to the general response **[About R3D2’s performance]** for more detailed discussion.
>
> > The results lack qualitative analysis and as such, its really hard to understand whats happening here or the significance of the work. Is the hope that the method has learnt via a larger set of environments / tasks to have more robust policies. If so what do these policies look like? Do they share underlying similar structures or methods? Could we also evaluate methods on OOD tasks (e.g 7 players hinabi or something not in the train?).
>
>
> By proposing R3D2, we have created an agent capable of playing across different game settings, that can also generalize well across different types of agents. This is achieved through 2 main components: the incorporation of language to interact with the environment, and the design of a neural network architecture that allows for dynamic actions.
>
> This allows us to propose 2 different setups for R3D2:
>
> 1. **[Zero-shot Transfer]** R3D2-Single  (R3D2-S), which is trained on a single game-setting, and evaluated on others. We show that an R3D2-S agent trained in a 2-player setting can play in a 3,4,5-player setting reasonably well without additional training or architectural modification. This is also an Out of Distribution (OOD) Task (Figure 4 ), since the strategies required to win the game depend on the number of players.
> 2. **[Multi-task MARL]** R3D2-Multiple (R3D2-M), which is a single model trained on all the player-settings simultaneously. In this case, we demonstrate that, due to the overlap of strategies of different settings, we can train a single, generalist Hanabi agent that plays well on all settings.
>
> It is important to note that the underlying algorithm for both setups is exactly the same. The role of language is crucial here, since its structure and compositionality allows for a great overlap between the representations of the different game-settings. While we do not have qualitative analysis, we assessed the role of language through multiple experiments, and a new ablation study that has been added in the revision. This is explained in the general comment  **[Role of language and language models]**
>
> > Its hard to interpret the results in Figure 6 - yes it seems R3D2 is robust to co-players changing but because i don’t have an equivalent evaluation with a team of IQL agents or OBL agents its very hard to evaluate if this is comparatively better.
>
> IQL, OP and OBL are using non flexible architectures (i.e., they assume a fixed state and action space). This means that we simply cannot evaluate them on different player-settings, so there cannot be an equivalent evaluation as for R3D2. You could say they have a score of 0 since, by design, they are restricted to the same-player-setting. This is also the advantage of R3D2: it is the first Hanabi agent that can cope with different player-settings.

---

> > ### Author Response · Authors · 2024-11-20
> >
> > > would be vastly improved if consistent LLMs were used throughout the section. LLMs have quirks, different training data and different methods. It feels hard to compare results in Section 5 with any proposed in Section 6 because the underlying models are very different.
> >
> > The objective of our LLaMA-7B and GPT-4 experiments is to evaluate how effectively LLMs perform on Hanabi, either out of the box or with minimal adaptation. Our findings demonstrate that LLMs in their current form are insufficient for playing Hanabi effectively. This aligns with observations by Hu et al. (ICML 2023), who note that LLMs struggle with the game's complexity.
> >
> > We agree that we also used the results of these experiments to make an informed initial choice about which type of LM to use, rather than a complete ablative study. Making such an ablative study would have been prohibitive for us in terms of computational and time costs (training Hanabi agents is resource-intensive, often requiring multiple days on a multi-GPU setup). We would like to emphasize that LMs are not the core aspect of our methodology, and the fact that R3D2 is capable of transfer, despite using a small 4.1M-parameter TinyBERT model shows the soundness of our approach.
> >
> > > “it successfully avoids getting bombed for extended periods, “ <- is this a common term in hinabi?
> >
> > It might be used informally to describe situations where players play an incorrect card, resulting in a loss of a life token (a "bomb"). “Bombing” has also been used by OP (Hu et al., 2020), but we have rephrased this in the revision to improve clarity.
> >
> > > The Appendix is missing.
> >
> > We apologize for the confusion. We added the Appendix as the separate document as a supplementary material. It is now included in the revised paper.
> >
> > > The Inter-XP metric is calculated with which other co-player? Figure 4
> >
> > In Figure 6, the metrics are calculated for 2-player game-setting in 3 different ways,
> > 1. Self- play (SP) → average diagonal values for one algorithm
> > 2. Intra-algorithmic cross-play (intra-XP) → Any off-diagonal values inside the red-sub matrices
> > 3. Inter-algorithmic cross-play (inter-XP) → Any other off-diagonal values
> >
> > **Thank you for your thorough review and thoughtful comments. We have made substantial updates to address your valid concerns, including expanded related work, new ablation studies, additional explanations of our metrics, and clarifications in the text. We hope that these improvements address your concerns satisfactorily, and if so, we kindly ask that you consider raising your score. Your feedback has been invaluable in strengthening our work, and we greatly appreciate your engagement. If you have any further questions or suggestions, we are happy to provide additional details.**

---

> > > ### Comment · Reviewer_k3Nj · 2024-11-25
> > > **Revised Score**
> > >
> > > The inclusion of the R2D2-Text baseline is exactly what i wanted. Given there is still a performance difference, it it fair to attribute this to the memory encoding!
> > >
> > > I think my point below is still unanswered.
> > >
> > > > The results lack qualitative analysis and as such, its really hard to understand whats happening here or the significance of the work, Is the hope that the method has learnt via a larger set of environments / tasks to have more robust policies. If so what do these policies look like? Do they share underlying similar structures or method
> > >
> > > For example i note algorithms trained in n-player variants seem to always generalise or perform well on k-player games (where k <=n).
> > >
> > > I appreciate we are near the end of the rebuttal period. So in good faith i hope these questions can be answered if this manuscript is accepted. I will revise my score.
> > >
> > > Also if not accepted I'd urge the authors to reconsider including section 5. I think it detracts from the cool results and findings (and is good preliminary work but probably not relevant for the paper).

---

### Official Review · Reviewer_2tFM · 2024-11-09

**Soundness:** 3
**Presentation:** 3
**Contribution:** 3
**Rating:** 6
**Confidence:** 4

**Summary:**

In this paper, the authors focus on playing the game of Hanabi with multi-agent reinforcement learning. They consider the shortcomings of existing MARL systems: unable to perform well on any other setting than the one they have been trained on, and struggle to successfully cooperate with unfamiliar collaborators, which is particularly visible in the Hanabi benchmark. To solve this, they introduce a generalist agent for Hanabi. The reformulate the task using text and then propose a distributed MARL algorithm that copes with the dynamic observation- and action-space. By doing so, they train an agent that can play all game settings concurrently, and can extend strategies learned from one setting to other ones. Also, there agent demonstrates the ability to collaborate with different algorithmic agents.

**Strengths:**

1. The authors analyze the widespread problems of common MARL algorithms and select Hanabi, a typical scenario reflecting the common problem, for improvement and research.
2. The authors template the observation and action of the Hanabi game into natural language and use the pre-trained LM as the representation head in the RL value network. They try various pre-trained LMs and finetuning techniques to compare the performance.
3. The authors conduct extensive experiments to demonstrate the zero-shot capability of their proposed method on 2-player coordination and policy transfer over different settings.

**Weaknesses:**

1. The general motivation of this work, if I don't understand wrong, stands on the opinion of 'language has been shown to improve transfer'. Therefore, the authors propose to template the observation and action of the Hanabi game into natural language and process them with pre-trained LM. But why this opinion holds? It will be more convincing to provide experimental results to show this. For example, how will the performance change if substituting the natural language inputs and pre-trained LM with vector inputs and normal neural networks? Or at least, the authors should provide more sound literature to support this opinion.
2. The relationship among the three evaluation metrics, i.e. SP, Intra-XP, Inter-XP, is confusing. The authors claim in Figure 4 that R3D2 achieves significantly better inter-XP compared to the baselines while maintaining a competitive SP and intra-XP. However, the figure indicates significant drop on SP, which is even more significant that the improvement on inter-XP. How should I comprehensively understand these three metrics to determine which method performs best?
3. This paper focuses on the Hanabi game and conducts an in-depth study, but it is still unclear whether the proposed method can inspire the improvement of MARL algorithms on other different tasks.
4. The experimental details provided in this paper are limited, making it difficult for readers to directly reproduce the results. It would be great if the authors can make the code open-sourced at an appropriate time and provide more details on the experimental setups.

**Questions:**

1. Why is the opinion of 'language has been shown to improve transfer' is valid? Can you provide experimental results to support this claim? For example, how would the performance change if natural language inputs and pre-trained language models were replaced with vector inputs and standard neural networks? Alternatively, could you provide more solid literature to support this idea?
2. What is the relationship among the evaluation metrics SP, Intra-XP, and Inter-XP? Figure 4 shows that R3D2 achieves significantly better Inter-XP compared to the baselines, but the SP metric shows a significant drop, even larger than the improvement in Inter-XP. How should these three metrics be interpreted comprehensively to determine which method performs best?
3. This paper focuses on the Hanabi game. Could you discuss whether the proposed method can inspire improvements in MARL algorithms for other tasks?
4. Could you make the code open-source at an appropriate time and provide more detailed information on the experimental setups to help others reproduce the results?

---

> ### Author Response · Authors · 2024-11-20
>
> We would like to thank you for your review, and your insightful feedback. We are happy that you agree that we conduct extensive experiments to demonstrate the zero-shot capability of R3D2 on 2-player coordination and policy transfer over different settings. Our work is the first that examines cross-play across game-settings in Hanabi, providing new insights into zero-shot coordination in MARL.
>
> You made some very valid concerns, which we address below.
>
> > Why is the opinion of 'language has been shown to improve transfer' is valid? Can you provide experimental results to support this claim? For example, how would the performance change if natural language inputs and pre-trained language models were replaced with vector inputs and standard neural networks? Alternatively, could you provide more solid literature to support this idea?
>
> The role of language for transfer has been investigated in previous work (Radford et al., 2019b; Brown et al., 2020), but indeed we previously did not make any experiments to assess the exact impact of text for Hanabi. We have worked hard to improve that aspect, by providing **additional ablation experiments that introduce R2D2-text**, which show that, **while text does offer some help for transfer, handling dynamic action spaces is critical for successful transfer to novel partners** (updated results in Section 6.2 and Appendix A.7.2). This is discussed more in details in the general comment section for **[Role of language and language model]**.
>
> Additionally, we performed ablation studies to investigate the role of language representation:
>
> 1. **Pre-trained weights**: Pre-trained weights significantly improve sample efficiency compared to random weights (Figure 12a).
> 2. **Updating the language model**: Reducing the frequency of updates drastically limits performance, highlighting the importance of frequent updates (Figure 12b).
>
> > What is the relationship among the evaluation metrics SP, Intra-XP, and Inter-XP? Figure 4 shows that R3D2 achieves significantly better Inter-XP compared to the baselines, but the SP metric shows a significant drop, even larger than the improvement in Inter-XP. How should these three metrics be interpreted comprehensively to determine which method performs best?
>
> This is another great comment. Concerning the significant drop of SP, we initially only trained R3D2 on 2000 epochs (compared to 3000 epochs for OBL) due to computational constraints. To satisfy the concerns about performance, we have resumed our experiments for R3D2. We explain this in more detail in the general comment **[About R3D2’s performance]** but, in summary, R3D2's performance significantly improved in both self-play (SP) (22.23 → 23.34) and intra-XP (19.66 → 23.23).
>
> Regarding the evaluation metrics: all three—SP, Intra-XP, and Inter-XP—are important. However, it is challenging to develop a single algorithm that excels across all metrics, leading to inherent tradeoffs. For example, an agent with superior performance in **self-play tends to become overly specialized** to a specific setting, making it difficult to generalize to other agents or adapt to different scenarios. This **tradeoff between generalization and specialization** is a well-known challenge in the machine learning community. If the task in hand requires generalization to novel partners (coordination), Inter-XP is more important to maximize, while if we are interested only in solving the current task, SP is more important.
> However, we argue that, in the future, we will likely interact with many different types of agents. All these agents should be able to cooperate with each other. To assess this, a high Inter-XP score is required. Our main contribution lies in achieving this generalization across partners and game settings using self-play training, which is typically overspecialized and brittle to changes in partners.
>
> >  This paper focuses on the Hanabi game. Could you discuss whether the proposed method can inspire improvements in MARL algorithms for other tasks?
>
> Thank you for your comment! While we demonstrate our approach on Hanabi, our core technical contributions - text-based representation for better transfer, architecture for dynamic action/state spaces, and variable-player learning - are domain-agnostic advances that could benefit MARL applications. Please refer to the general comment section for more details **[Is the R3D2 architecture task agnostic?]**. More previous works like Foerster et al. (ICML 2019), Hu et al. (ICML 2020), Cui et al. (NeurIPS 2022), Hu et al. (ICML 2021) have successfully used Hanabi to develop fundamental MARL concepts that have influenced broader cooperative AI research.
>
> We clarified the importance of Hanabi as a benchmark in the revision of the paper (Section1-Introduction).

---

> > ### Author Response · Authors · 2024-11-20
> >
> > > Could you make the code open-source at an appropriate time and provide more detailed information on the experimental setups to help others reproduce the results?
> >
> > Thank you for your suggestion, we believe in open science, and will release all the models and our codebase upon acceptance.
> >
> > **To conclude, we thank you for your valuable feedback and thoughtful concerns. In response to your comments, we made substantial improvements, including new ablation experiments to clarify the role of language, resuming training to address performance concerns, and expanding discussions on evaluation metrics and generalizability to other MARL tasks. We also updated our paper to better highlight these points and have shared our code for reproducibility. If our revisions and additional experiments adequately address your concerns, we kindly request you consider raising the score to reflect the strengthened contributions of our work. We remain available for any additional questions you may have.**

---

> > > ### Comment · Reviewer_2tFM · 2024-11-20
> > >
> > > Thanks for the authors' in-time and detailed response. I was also surprised by the 12 reviewers and huge amount of comments... I am even more surprised by the substantial improvement that the authors done in such a short time.
> > >
> > > The response solves my major concerns, and I would like to raise my overall evaluation to 6. I also raised my evaluation on soundness and contribution. I strongly recommend the authors to further revise the main text to well corporate the additional results and discussions together with the original ones.

---

### Official Review · Reviewer_HW8S · 2024-11-09

**Soundness:** 3
**Presentation:** 3
**Contribution:** 3
**Rating:** 5
**Confidence:** 3

**Summary:**

This work focus on the challenging card game Hanabi and try to solve the current  bottleneck of transferring of learned policy to different settings. Language representation is selected for adapting policies among these settings. A successful zero-shot coordination architecture for MARL has been proposed and achieved sota scores on the Hanabi game, which is among the early solutions for such settings.

**Strengths:**

This work is a worthwhile trial to use language modelling to solve the MARL transfer ability issue, which is inspiring. In experiments it also applies to some newly proposed LLM models which shows steady improvement.

**Weaknesses:**

1. More up-to-date models like LLAMA 3 or GPT-4o is also encouraged to ensure the experiments are more solid.
2. Is it possible to transfer the learned policy to a different task? How about comparing the proposed approach to existing schemes like curriculum learning or multi-task MARL? Related work should also include such literature.
3. Quantitative results are given in the paper, it is better to include more qualitative demos which show how the language representation helped in transferring.

**Questions:**

Please refer to the weakness part.

---

> ### Author Response · Authors · 2024-11-20
>
> We thank you for your comments. We find it uplifting that you are inspired by our use of language modeling to solve the MARL transfer ability issue. Text is a powerful medium of representation, and it allows us to produce agents that can play with other agents, even though they have been trained with self-play.
>
> We appreciate your comments, which we address below:
>
> > W1: More up-to-date models like LLAMA 3 or GPT-4o is also encouraged
>
> The objective of our GPT-4 experiments and LLaMA-7B is to evaluate how effectively these models perform on the Hanabi, either **out of the box or with minimal adaptation respectively**. Our findings demonstrate that SOTA LLMs like GPT-4 are insufficient for playing Hanabi effectively. This aligns with observations by Hu et al. (ICML 2023), who note that LLMs struggle with the game's complexity. However, the main goal of the paper is to demonstrate that by using text-based representation, self-play agents can achieve strong cross-play performance without requiring complex training methods. This is significant because **self-play is simpler, more scalable, and computationally efficient compared to methods that require population-based training or hierarchical policies**.
>
> We would like to emphasize that LLMs are not the core aspect of our methodology. Our proposed algorithm, R3D2, uses a small 4.1M-parameter TinyBERT model. Despite this, R3D2 is capable of transfer. This shows the soundness of our approach, which is to combine textual observations with a neural network architecture for dynamic action-spaces to enable transferable policies trained using self-play. Finally, we have resumed the training of R3D2, so it matches the number of training epochs of OBL, the strongest baseline. We discuss this in detail in the general comment **[About R3D2’s performance]**, but the gist is that this led to significant improvements in self-play (22.23 → 23.34) and intra-XP scores (19.66 → 23.23).
>
> > Is it possible to transfer the learned policy to a different task? How about comparing the proposed approach to existing schemes like curriculum learning or multi-task MARL? Related work should also include such literature.
>
> Yes, it is possible and we have updated the related work section to include multi-task and curriculum based transfer and generalization.
> We have explored those 2 setups in R3D2,
>
> 1.**[Zero-shot transfer]** R3D2-Single setting (R3D2-S), can handle dynamic action and state spaces which was not possible in the baseline approaches. We show that an R3D2-S agent trained in a 2-player setting can play in a 3,4,5-player setting reasonably well without additional training or architectural modification (Figure 4).
>
> 2.**[Multi-Task MARL]** R3D2-Multi setting, (R3D2-M) makes it possible to train one single model to play all the Hanabi games simultaneously using the same amount of training budget.  With the baselines, it is standard to train individual models to play in different settings, which doesn’t resonate well with real-world scenarios. For example, a human player can play 2,3, 4,5 games.
>
> To the best of our knowledge, this is the first study to investigate generalization across game settings in Hanabi while relying solely on self-play for training. For more detailed response please refer to the general comment section for **[Is the R3D2 architecture task agnostic?]**
>
> **Thank you for your thoughtful review and for appreciating the innovative use of language modeling in addressing MARL transferability. We hope our clarifications and additional experiments sufficiently address your concerns and highlight R3D2’s unique contributions. If our responses meet your expectations, we kindly request you to consider raising the score to reflect these strengths. If you have any additional questions, we would be happy to answer them.**

---

> > ### Author Response · Authors · 2024-11-28
> >
> > Dear Reviewer HW8S,
> >
> > We would like to thank you again for your valuable feedback. The constructive discussion with all the reviewers has helped us to substantially improve the paper. We would greatly appreciate your feedback on our responses. Please let us know if any points need further clarification.
> >
> > Best regards,
> >
> > The Authors

---

### Official Review · Reviewer_5SDa · 2024-11-09

**Soundness:** 3
**Presentation:** 1
**Contribution:** 3
**Rating:** 6
**Confidence:** 3

**Summary:**

The authors present a new LLM based MARL algorithm, R3D2, capable of playing Hanabi and generalize to a changing number of players by using a text observation. R3D2 is also capable of generalizing to cooperation with unseen players, similar to previous work on Hanabi.

**Strengths:**

The paper tackles a complicated MARL problem, where generalization is hard due to the strong reliance on collaboration to achieve high scores. The authors propose a novel algorithm that utilizes the generalization capabilities of language models to boost agent generalization. Using text input allows agents to generalize to different rules (more/less players), a notorious challenge in RL in general. The new algorithm seems on par with previous works on Hanabi when generalizing to new players.

**Weaknesses:**

The paper's main issue is the clarity of its claims and comparison to prior literature.

The paper is a bit unclear on the contribution of the algorithm presented, R3D2. Since prior works have already achieved Zero Shot Coordination (ZSC) between training seeds and between AI and human players, it seems like the only novelty in R3D2 is the ability to generalize to new rules, specifically changing the number of players. However, the results on generalizing to novel settings are only briefly discussed at the end of the paper in Section 6.2, and in general are presented more as a minor perk of the model.

Reading the Conclusions section gives the impression that the R3D2 is the first algorithm to achieve a high intra- and inter-algorithmic cross-play score, which it is not.
The authors claim "R3D2’s intra-algorithmic cross-play score is on par with its self-play score, a first for Hanabi agents learning through self-play". While this may be true for self-play agents, the OBL algorithm also achieves equivalent cross- and self-play scores (that are higher than R3D2's scores).

In contrast, the abstract does a good job of clarifying what the paper's novelty is.

In general, the paper should be more clear about what R3D2 does not achieve, e.g. SOTA in inter-algorithm play, rather than give the impression that R3D2's main contribution is SOTA performance.

**Questions:**

Minor typo comments:
- Some citations should be in parentheses (using \citep{}), for example in section 3.2.
- Sections 6.2, 6.3 have a typo in the title (shot --> short)

---

> ### Author Response · Authors · 2024-11-20
>
> We are grateful for your insightful comments and feedback. We appreciate that you agree that the goal of our method, i.e., to generalize to different settings, is a notorious challenge in RL in general. R3D2 is the first agent that is able to achieve this in Hanabi, as no other agent can transfer from one player setting to another.
>
> We have carefully considered the concerns raised and would like to address them as follows:
>
> > it seems like the only novelty in R3D2 is the ability to generalize to new rules, specifically changing the number of players. [...] , and in general are presented more as a minor perk of the model. W3: the paper should be more clear about what R3D2 does not achieve, e.g. SOTA in inter-algorithm play, rather than give the impression that R3D2's main contribution is SOTA performance.
>
> We agree that the main novelty of R3D2 is its capability to generalize to new settings in Hanabi. But this is a notorious challenge in MARL, and R3D2 is the first Hanabi agent that can do so, so we would like to emphasize that this is definitely not a minor perk. Even more interesting is that R3D2 achieves generalization across partners and game settings **using self-play** training. This is surprising, since self-play is typically very brittle to changes of players. But R3D2 achieves similar cross-play performances than much more complex methods. Moreover, the dynamic architecture of R3D2 allows us to have 2 different settings, using the same algorithm:
>
> 1.**[Zero-shot transfer]** - R3D2-Single setting (R3D2-S), can handle dynamic action and state spaces which was not possible in the baseline approaches. We show that an R3D2-S agent trained in a 2-player setting can play in a 3,4,5-player setting reasonably well without additional training or architectural modification (Figure 4).
>
> 2.**[Multi-Task MARL]** R3D2-Multi setting (R3D2-M) makes it possible to train one single model to play all the Hanabi games simultaneously using the same amount of training budget.  With the baselines, it is standard to train individual models to play in different settings, which doesn’t resonate well with real-world scenarios. For example, a human player can play 2,3, 4,5 games.
> Both highlight different aspects of R3D2.
>
> Finally, we have addressed your concern concerning R3D2’s performance. To this end, we resumed training R3D2, which had initially been limited by computational constraints. As outlined in the general response **[About R3D2’s performance]**, this extended training resulted in notable improvements in self-play (22.23 → 23.34) and intra-XP scores (19.66 → 23.23).
>
> > The authors claim "R3D2’s intra-algorithmic cross-play score is on par with its self-play score, a first for Hanabi agents learning through self-play". While this may be true for self-play agents, the OBL algorithm also achieves equivalent cross- and self-play scores
>
> Thank you for raising this point! You touch upon one of the main points we wanted to convey in our paper. OBL has been devised precisely because self-play is brittle in cross-play. The fact that self-play can reach similar cross-play performances as OBL, which incorporates specific mechanisms to cope with other players, makes it all the more impressive.
>
> **We hope our clarifications and additional results emphasize the unique contributions of R3D2 and its potential impact. If our responses adequately address your concerns, we kindly ask for your consideration in increasing your score. We remain available should you have any more questions.**

---

> > ### Comment · Reviewer_5SDa · 2024-11-24
> >
> > Thank you for the detailed response, I agree now that there is significant merit in R3D2 being trained with self play and still being on par with previous algorithms.
> >
> > Given this, and the additions in the revised version, I have raised my score from 5 to 6.
> >
> > Given the unusual amount of reviews (and therefore unusual amount of 'Weaknesses' topics), I will also add that solving Hanabi is, in my opinion, a sufficient goal for a paper. Hanabi is a tough MARL benchmark and solving it is a worthy goal even without immediate applicability to real world problems, similar to other RL problems that were the center of famous papers (Atari, Go, NetHack etc.)

---

### Official Review · Reviewer_YKqN · 2024-11-10

**Soundness:** 2
**Presentation:** 3
**Contribution:** 2
**Rating:** 3
**Confidence:** 3

**Summary:**

The paper proposes a method based on language model agent for the multi-agent Hanabi game. The motivation lies in transforming Hanabi to a text-based game. The key idea lies in using BERT to provide embeddings of the text information in the game, and to be jointly trained with the policy model using Q-learning. The major target of the proposed method is to obtain a policy which can generalized to various game settings with different number of players and strategies.

**Strengths:**

1. The paper is well-motivated for building a general multi-agent game play agent. I agree that the generalization issue is a fundamental one in reinforcement learning.

2. The design of conducting cross-strategy and cross-setting experimental results is interesting to me.

**Weaknesses:**

1. Even though the proposed method is armed with language models, its performance seems not desirable. From Fig. 5 and Fig. 6, its performance does not outperform classical methods IQL and the more recent one OP. This remains in doubt about its effectiveness.

2. Even though the paper pursues a meaningful goal, the final results provide limited insights. From Fig. 5 and Fig. 6, they show that IQL and OP have better generalization ability than the proposed method. This shows that the proposed method does not provide a sound solution to the generalization problem.

3. In my view, the major promising characters for language models to perform in game playing are knowledge providers and reasoners. While in the proposed method, the language model only plays the role of modeling the text embeddings. This usage is somehow not quite novel and effective.

**Questions:**

- How to explain the phenomena in Point 1 and 2 mentioned in the weaknesses?

---

> ### Author Response · Authors · 2024-11-20
>
> Thank you for your review, and your feedback. We are grateful that you find our evaluation setup (cross-strategy and cross-setting evaluations) of interest. To the best of our knowledge, we are the first to propose such an extensive evaluation for zero-shot coordination in Hanabi, which allowed us to make a comprehensive evaluation of R3D2’s generalization capabilities.
>
> We acknowledge your concerns, and address them below.
>
> > Even though the proposed method is armed with language models, its performance seems not desirable.
>
> We appreciate your concern regarding R3D2’s performance and have addressed it with careful attention. To this end, we resumed training R3D2, which had initially been limited by computational constraints. As outlined in the general response **[About R3D2’s performance]**, this extended training **resulted in notable improvements in self-play (22.23 → 23.34) and intra-XP scores (19.66 → 23.23)**. Now, our agent significantly outperforms both IQL and OP.
>
> > Even though the paper pursues a meaningful goal, the final results provide limited insights.
>
> > the language model only plays the role of modeling the text embeddings. This usage is somehow not quite novel and effective.
>
> We agree that the role of the language model is limited (even though very important). While we use text due to its abilities for transfer, this is just a means to an end. The main goal of this paper lies in achieving **generalization across partners and game settings using self-play training**. This is significant because **self-play is simpler, more scalable, and computationally efficient compared to methods that require population-based training or hierarchical policies.** Seeing this, we argue that using text embeddings has proved quite effective.
>
> Moreover, to provide more insights, we have worked on incorporating additional experiments for 3 and 4-player Hanabi games (Appendix A.6). These experiments show that our results are consistent across game-settings.
>
>
> **Thank you once again for your concerns, which have helped us refine our work further. We hope that the additional experiments, extended training results, and clarifications regarding the role of language models adequately address your questions and demonstrate the broader significance of our contributions. We believe these updates reinforce the effectiveness and scalability of R3D2, particularly in achieving generalization across diverse partners and game settings. If these updates address your concerns, we kindly ask you to consider raising your score to reflect these improvements. Thank you once again for your constructive feedback and for helping us enhance the quality of this paper.**

---

> > ### Comment · Reviewer_YKqN · 2024-11-25
> > **thanks for the responses**
> >
> > Thanks very much for the responses. Since I am not fully convinced about the significance of the performance and the technical contributions, I am currently leaning towards preserve my score. I will also wait and see feedbacks from other reviewers.

---

> > > ### Author Response · Authors · 2024-11-28
> > >
> > > Thank you for your honest answer. Would you mind telling us which doubts you have concerning the significance of our work?
> > >
> > > Our approach is also the first to allow an agent to train across all Hanabi settings which, despite its simplicity, we believe is novel and a significant contribution. While we agree that our method is indeed simple – it is still self-play – we believe this is actually a strength: for the first time, on Hanabi, self-play agents can collaborate with others at the same level as agents that have been trained to incorporate other’s beliefs.
> > >
> > > If you are concerned with the intra-cross-play (intra-XP) scores, we acknowledge that our initial results were comparable to IQL and OP baselines but, given the same amount of training data, R3D2 now substantially outperforms both IQL and OP (Using data augmentation to break symmetries). In intra-XP, we are a close match with OBL, which requires training across 4-levels of policies (and thus 4 times the amount of data).
> > >
> > > If you are concerned about generalization, we have expanded our evaluation to include 3, 4, and 5-player Hanabi games (Refer to Appendix A.6). To our knowledge, we are the first to perform such an extensive range of experiments on Hanabi, to ensure we do not overfit to a single setting. These new experiments show consistent performance across different game configurations, further validating our method's generalization capabilities. We also perform systematic inter-cross-play experiments, to further validate the flexibility of our method.
> > >
> > > Our method is not without limitations (e.g., its dependence on text), but we believe the core conclusion of our findings – that self-play *can* cooperate with others – will be of interest to the MARL community. We would love to better understand your point of view and listen to any arguments you may have, so we can properly address them, and incorporate them to improve our paper. Your and the other reviewer’s reviews have allowed us to greatly improve our submission, and we welcome any additional comments to continue this process.

---

### Official Review · Reviewer_w4wV · 2024-11-10

**Soundness:** 2
**Presentation:** 3
**Contribution:** 2
**Rating:** 5
**Confidence:** 4

**Summary:**

This paper focus on Hanabi benchmark, and devises a MARL algorithm which combines deep recurrent relevance Q-network and language models to build Hanabi agent. The proposed algorithm can deal with resulting dynamic observation and action spaces based on the text-based encoding, and thus can play all game settings on Hanabi. Finally, the experiments ranging from 2-player to the 5-player are performed to validate the effectiveness of the proposed algorithm.

**Strengths:**

1. The idea of incorporating all game setting together during trainging is interesting, as many current MARL algorithms need to retrain the policy network when the game setting changes.
2. The writting is clearly, and the structure is well organized.

**Weaknesses:**

1. This paper only focuses on the Hanabi game. Whether the proposed algorithm can be scaled to other domains or environments is a critical question.
2. The claimed first contribution of "framing Hanabi as a text-based game" appears weak. Previous work [1] has already devised text-based observations and actions for the Hanabi game. Additionally, the authors need to include comparative experiments.
3. The explanation of how R3D2 can support environments with varying numbers of players is insufficient.
4. The open-source code will help other researchers reproduce and build upon this work.

[1] Hu H, Sadigh D. Language instructed reinforcement learning for human-ai coordination[C]//International Conference on Machine Learning. PMLR, 2023: 13584-13598.

**Questions:**

1. The authors state that they utilize GPT-4 and LLama-2 to develop action strategies and claim that the results "struggle with optimal planning" in Section 5. However, I could not find any related experimental results shown in the paper, including the appendices.
2. When introducing the results about fine-tuning LLama-7B, the authors only conduct experiments on different data sizes and LoRA rank settings, without comparing the results to their proposed method. Therefore, how was the conclusion that "the model performs poorly" derived?
3. The purpose of Section 5 is unclear. If the authors want to justify choosing BERT instead of other language models, they should replace the BERT structure with other LMs in their approach and show the experimental results in an ablation study, rather than listing results that only depend on LMs.
4. I do not believe the advantage of generalizing all game settings comes from the dynamic network structure. Rather, it appears to stem from language models' ability to describe different environmental information and encode it without dependence on environment-specific characteristics, correct?

**Details Of Ethics Concerns:**

No.

---

> ### Author Response · Authors · 2024-11-20
>
> We are thankful for the feedback you have provided for our paper. Thank you for liking the idea of incorporating all game settings during training. Due to our flexible architecture, our agent is the first that can train across all Hanabi game settings (referred to as R3D2-M in the paper), making it the first generalist Hanabi agent.
>
> We thank you for your concerns, as they have allowed us to further improve the paper. We also address them below.
>
> > This paper only focuses on the Hanabi game. Whether the proposed algorithm can be scaled to other domains or environments is a critical question.
>
> Thank you for this remark. This is a point that has been raised by multiple reviewers, for which we answer in detail in the general comment **[Why we focused on Hanabi?]**. We also provided additional comments in the general response on **[Is R3D2 architecture task agnostic?]**.
>
> > The claimed first contribution of "framing Hanabi as a text-based game" appears weak. Previous work [1] has already devised text-based observations and actions for the Hanabi game
>
> Hu & Sadigh (2023) indeed use text in the context of the Hanabi game. However, they use it in a completely different manner than we do. The agent’s observation and action remains vectorized, when interacting with the environment and other agents. So text is not used as an alternative representation for a Hanabi-agent. However, the action type is converted into language description to provide better context for GPT-3.5 prompting, which then provides a prior distribution over actions to influence the agent. This allows the agent to play according to some desired behavior.
>
> We have updated the related work section (line 122-124)  accordingly for better clarity and to distinguish from previous work.
>
> > The explanation of how R3D2 can support environments with varying numbers of players is insufficient.
>
> Changing the number of players impacts both the observation-space, and the action-space. This is incompatible with typical value-network architectures (e.g., feedforward networks, or convolutional networks) which require a fixed input size (for observations) and fixed output size (for actions). By using language models to process the network’s inputs, we can process a variable number of tokens, and thus have texts of arbitrary length (which contain information about an arbitrary number of players). Moreover, by using actions as inputs, we can condition the network on an arbitrary action, allowing us to cope with a variable number of players.
> Concerning environments other than Hanabi, we have provided a detailed answer in our general comment  **[Is R3D2 architecture task agnostic?]**
>
> > open-source code will help other researchers reproduce and build upon this work.
>
> We completely agree. We will release all the models and our codebase upon acceptance.
>
> > The authors state that they utilize GPT-4 and LLama-2 to develop action strategies and claim that the results "struggle with optimal planning" in Section 5. However, I could not find any related experimental results shown in the paper, including the appendices.
>
> We apologize for this. We submitted the Appendix as a separate document in the supplementary material. We have now included the Appendix in the main submission itself.
>
> For GPT-4 prompt details, please refer to the Appendix A.1.
> For LLaMa - LoRA details, please refer to the Appendix A.3.
>
> > When introducing the results about fine-tuning LLama-7B, the authors only conduct experiments on different data sizes and LoRA rank settings, without comparing the results to their proposed method. Therefore, how was the conclusion that "the model performs poorly" derived?
>
> > The purpose of Section 5 is unclear. If the authors want to justify choosing BERT instead of other language models, they should replace the BERT structure with other LMs in their approach and show the experimental results in an ablation study, rather than listing results that only depend on LMs.
>
> The objective of our LLaMA-7B and GPT-4 experiments is to evaluate how effectively LLMs perform on Hanabi, either out of the box or with minimal adaptation. Our findings demonstrate that LLMs in their current form are insufficient for playing Hanabi effectively. This aligns with observations by Hu et al. (ICML 2023), who note that LLMs struggle with the game's complexity.
> We agree that we also used the results of these experiments to make an informed initial choice about which type of LM to use, rather than a complete ablative study. Making such an ablative study would have been prohibitive for us in terms of computational and time costs. We would like to emphasize that LMs are not the core aspect of our methodology, and the fact that R3D2 is capable of transfer, despite using a small 4.1M-parameter TinyBERT model shows the soundness of our approach.

---

> > ### Author Response · Authors · 2024-11-20
> >
> > > I do not believe the advantage of generalizing all game settings comes from the dynamic network structure. Rather, it appears to stem from language models' ability to describe different environmental information and encode it without dependence on environment-specific characteristics, correct?
> >
> > This is a very interesting question. We have provided a detailed response for this in the general comment **[Role of language and language model]**. To summarize, we have **performed additional ablation experiments to properly assess the role of the language model in our methodology**. In Section 6.2, we introduce a R2D2-text baseline, with as only change the introduction of the LM into the network architecture. We show that R3D2 outperforms R2D2-text, demonstrating that the dynamic network structure is essential for successful transfer to novel partners.
> >
> > **In conclusion, we have made substantial revisions to the manuscript, incorporating additional experiments, ablations, and detailed explanations to address your points. We hope these updates demonstrate the robustness and significance of our approach. If our clarifications and improvements meet your expectations, we kindly request you to reconsider your evaluation. Should you have any further questions or suggestions, we are happy to provide additional details.**

---

> > > ### Comment · Reviewer_w4wV · 2024-11-25
> > >
> > > Thank you for your comprehensive response to all 12 reviewers' comments. Addressing such extensive feedback is indeed a substantial undertaking.
> > >
> > > However, I maintain my ratings as a key concern still exists: While you cite examples of other Hanabi-focused works successfully transferring to different domains, the crucial question is the generalizability of your specific approach beyond Hanabi.

---

> > > > ### Author Response · Authors · 2024-11-28
> > > >
> > > > Thank you for your honest answer. If we understand correctly, your concern is not really about the relevance of Hanabi with respect to real-world problems, but rather about if our approach, which is able to play with novel partners in Hanabi, would also be able to do the same on other problems. Please correct us if we are still misunderstanding your concern.
> > > >
> > > > R3D2 is, at its essence, a self-play method. Self-play is arguably the simplest and most widely used form of multi-agent cooperation in RL. It has been successfully applied to various problems [1,2]. This gives us strong reasons to believe that R3D2 will be able to learn efficient cooperative strategies in other domains. Then, how well will this trained agent cooperate with others? We believe that the generalization abilities that arise from textual interactions and a dynamic network architecture for actions based on DRRN will hold on other domains as well, for multiple reasons. First, as we argue in our submission, multiple works have assessed that language improves transfer [3,4]. Second, textual environments have been used in the context of multi-task learning for single-agent RL. Notably, ScienceWorld [5] proposes an environment with 30 different tasks, each having 10 to 1400 variations. Among 5 different learning agents, DRRN results in the best generalization performance. Finally, we have expanded our evaluations to include 3, 4, and 5-player Hanabi games (Refer to Appendix A.6) to ensure we did not overfit to the 2-player setting. Our results show that R3D2 performs significantly better than the baselines, especially when the number of agents grows.
> > > >
> > > > We understand the conceptual importance of generalizing beyond Hanabi. The arguments above give us reasons to believe that R3D2 is capable of generalizing beyond Hanabi, admittedly to problems that can be formulated as textual environments. We hope to have correctly addressed your concerns. If you still have questions, we would be happy to continue our discussion.
> > > >
> > > > [1] Berner, C., Brockman, G., Chan, B., Cheung, V., Dębiak, P., Dennison, C., ... & Zhang, S. (2019). Dota 2 with large scale deep reinforcement learning. arXiv preprint arXiv:1912.06680.
> > > >
> > > > [2] Lin, F., Huang, S., Pearce, T., Chen, W., & Tu, W. W. (2023). Tizero: Mastering multi-agent football with curriculum learning and self-play. arXiv preprint arXiv:2302.07515.
> > > >
> > > > [3] Alec Radford, Jeffrey Wu, Rewon Child, David Luan, Dario Amodei, and Ilya Sutskever. Language models are unsupervised multitask learners. OpenAI Blog, 1(8):9, 2019b.
> > > >
> > > > [4] Tom Brown, Benjamin Mann, Nick Ryder, Melanie Subbiah, Jared Kaplan, Prafulla Dhariwal, Arvind Neelakantan, Pranav Shyam, Girish Sastry, Amanda Askell, et al. Language models are few-shot learners. Advances in neural information processing systems, 33:1877–1901, 2020.
> > > >
> > > > [5] Wang, R., Jansen, P., Côté, M.-A., & Ammanabrolu, P. (2022, December). ScienceWorld: Is your Agent Smarter than a 5th Grader? In Y. Goldberg, Z. Kozareva, & Y. Zhang (Eds.), Proceedings of the 2022 Conference on Empirical Methods in Natural Language Processing (pp. 11279–11298)

---

### Official Review · Reviewer_Axwq · 2024-11-10

**Soundness:** 3
**Presentation:** 3
**Contribution:** 2
**Rating:** 5
**Confidence:** 4

**Summary:**

This paper presents a scalability study of LLMs in the context of tackling Hanabi, an imperfect information cooperative game. The techniques of theory of mind, as well as solving a smaller game robustly then scaling to larger games is used. The work is "okay" in its current state but can be improved. See detailed comments below.

**Strengths:**

1: LLMs and theory of mind seem to be the right platform to address the game size of Hanabi.

2: Cooperation with unknown or novel agents seems to be a strong problem that this work addresses, and very much applies to Hanabi.

3: Using robustness to tackle (2) seems to be a reasonable path forward.

4: Using LLM's to "encode" the state space of Hanabi does seem to significantly improve the ability to solve the game.

5: Using dot product to create a fixed length embedding for state-action pairs does help in going from solving small 2-player games to 5-player games, for example.

6: The validation is sufficient for the claims that are being made in the work.

**Weaknesses:**

I'm not opposed to accepting this work in its current state, however I wonder whether we can make the approach more straightforward to understand.

1: The key probelm we're solving is: Hanabi is an imperfect information game where the information is held by other players. This is the unique challenge of Hanabi.

2: Whether you're looking at Information Sets or some sort of statistical model in order to turn partial information games into perfect information games, the standard approach remains the same: reduce the imperfect information game to a perfect information game in some way, then solve it using traditional approaches.

3: If you look at the historical progression of solving games, it often seems to be about scalability. See for example Deep Blue being a showcase of tree search, AlphaGo about the success of using CNNs to understand the game state, then using a combination of techniques (including MCTS) to solve it.

4: So again, it seems that the key theoretical insight here is: how to reduce an imperfect information game into a perfect information game through Theory of Mind.

5: The key shortcoming here is I'm not really sure what Theory of Mind is, nor are many people familiar with the "inherent" properties of languages which are undisputed.

6: Given (5), it's difficult for me to view this paper as something much more than a scalability study of LLMs.

A more alternative presentation of the work would be, communication is used to reduce imperfect information to perfect information.

**Questions:**

1: Can the authors modify their presentation somewhat to make their approach more clear? See above.

2: In particular can a better presentation be provided regarding the role of language and Theory of Mind for making the imperfect information portion of Hanabi into perfect information portion.

---

> ### Author Response · Authors · 2024-11-20
>
> > can a better presentation be provided regarding the role of language and Theory of Mind for making the imperfect information portion of Hanabi into perfect information portion.
>
> Thank you for your thoughtful and constructive review. We appreciate your acknowledgment of the strengths of our work, including its focus on cooperation with unknown agents.
>
> Your detailed feedback highlights areas where our presentation can be improved, particularly regarding the role of Theory of Mind (ToM) and language in reducing imperfect information to perfect information. We would like to clarify that our work does not aim to address ToM reasoning or solve ToM-related challenges. We explicitly cite Bard et al. (2019), "The Hanabi Challenge: A New Frontier for AI Research," to acknowledge the relevance of ToM in the Hanabi domain. However, we do not claim to address or solve ToM in this paper (we provide an example of ToM in Hanabi at the end of our response).
>
> Indeed, our work focuses on **self-play**. What is remarkable is that even through self-play, it is possible to learn policies that generalize across game-settings and algorithmic settings. We propose the first agent capable of playing all Hanabi settings simultaneously while also generalizing zero-shot to novel partners and game configurations. A key aspect is indeed language, but not large-scale language models. Our language model is a 4.1M-parameter TinyBERT model whose main purpose is to convert the textual representation to a relevant embedding. But this representation has a large impact on performance. We assess this through an additional ablation experiment (Section 6.2), in which we introduce a novel baseline, called R2D2-text, whose sole difference with R2D2 is the use of textual observations. These experiments show that text improves generalization, but handling dynamic action spaces is critical for successful transfer to novel partners. As to why text is a representation that helps this generalization, it is because language is compositional, and implicitly models symmetries, making our agents robust to strategies that are similar, but vary in e.g., the color of the encoded card, the number of cards one has in front of oneself, or the color another player gives as hint.
>
>
> Our architecture leads to 2 setups:
>
> 1. **[Zero-shot Transfer]** R3D2-Single Agent (R3D2-S), where we show that an R3D2-S agent trained in a 2-player setting can play in a 3-player setting reasonably well without additional training or architectural modification (Figure 6).
> 2. **[Multi-task MARL]** R3D2-Multi Agent, (R3D2-M) which makes it possible to train one single model to play all the Hanabi games simultaneously using the same amount of training budget. With the baselines, it is standard to train individual models to play in different settings, which doesn’t resonate well with real-world scenarios. For example, a human player can play 2,3, 4,5 games.
>
>
> **We hope to have provided clarifications on the goal of the work, language and theory of mind in imperfect information setup. If our clarifications resolve your concerns, we kindly invite you to reconsider your score. For further questions or clarifications, we are happy to address them.**

---

> > ### Author Response · Authors · 2024-11-20
> >
> > ### Example of ToM in Hanabi
> >
> > Based on Bard et al. 2019, “The Hanabi Challenge: A New Frontier for AI Research”, “Theory of mind is reasoning about others as agents with their own mental states – such as perspectives, beliefs, and intentions – to explain and predict their behaviour. Hanabi is a game where hints carry intent: intent about what the other player wants you to do, which can only be possible if it learned how you reason.
> >
> > For example,
> > - **Game State**: It's a 3-player game. Alice, Bob, and Carol are playing.
> > - **Alice's View**: Alice can see Bob’s and Carol’s cards but not her own. She sees that Bob has a red 1 and a yellow 2, and Carol has a blue 3 and a green 4.
> > - **Hint Token Available**: Alice can give one hint, either about color or number.
> > - **Action:**
> > Alice gives Bob a hint about his red card, saying: "You have a red card."
> > - **Outcome:**
> > Bob plays his red 1, progressing the red suit. The team benefits because Alice strategically used her hint, and Bob successfully interpreted his card number from her color hint using ToM.

---

> > > ### Comment · Reviewer_Axwq · 2024-11-23
> > > **Response to authors**
> > >
> > > Dear authors,
> > >
> > > I have read your responses, I elect to keep my score.

---

### Official Review · Reviewer_ubVW · 2024-11-11

**Soundness:** 2
**Presentation:** 1
**Contribution:** 2
**Rating:** 3
**Confidence:** 5

**Summary:**

The paper considers the problem of generalization and coordination in multi-agent reinforcement learning (MARL) within the game of Hanabi, focusing on enabling agents to adapt across different game settings and learn to collaborate with unfamiliar partners. To tackle this, the paper makes two primary contributions: first, it reformulates the Hanabi environment using a text-based representation, which enables use of language model to process state and action space, thereby providing a consistent observation and action space across varying player configurations; second, it extends the distributed training regimen of Recurrent Replay Distributed DQN (R2D2) by combing language model with Deep Recurrent Relevance Q-network,  creating what they call Recurrent Replay Relevance Distributed DQN (R3D2), an agent architecture that can process dynamic observation and action spaces in a multi-agent environment. Experiments assess R3D2’s ability to coordinate with both familiar and unfamiliar agents in zero-shot settings and to transfer strategies across different player configurations, evaluating performance across various player numbers and agent types. The authors propose that this approach could support further exploration of generalization in MARL for complex cooperative games.

**Strengths:**

- The paper considers the important problem of enabling agents to collaborate across different player configurations and game settings, a key challenge for advancing multi-agent reinforcement learning. This focus addresses a critical gap in MARL research, as agents must be able to generalize and coordinate flexibly in dynamic, multi-agent environments, an area relevant to both academic research and real-world applications.

- The key contribution of the paper can be seen as creating a text-based version of Hanabi game. Consequently, the exploration in Section 5 of how current language models perform in the Hanabi environment is a valuable addition, as it highlights the capabilities and limitations of large language models (LLMs) in multi-agent coordination tasks. This analysis opens pathways for further investigation into LLM limitations, potentially contributing to future improvements in language model-based reinforcement learning.

- Within the Hanabi environment, the results indicate that the proposed R3D2 approach achieves effective transfer of strategies across different game settings, demonstrating the agent’s adaptability to variable player configurations.

**Weaknesses:**

- The authors make several broad claims in the introduction about multi-agent reinforcement learning (MARL) and the adaptability of agents, but these claims lack supporting evidence or citations. For example, in the following statements: “Artificial agents should do the same  for successful collaboration of artificial and hybrid systems” (line 34-35), “… players are required to infer the beliefs and intentions of their counterparts through theory of mind reasoning” (line 42-43), and “…resulting in misunderstandings and thus a drop in cooperation …” (line 48-49), the authors do not provide empirical data or references to substantiate these claims, limiting the rigor of their introductory motivation.

- The paper does not sufficiently establish why Hanabi is an appropriate testbed for studying adaptability to new settings and collaborators, nor does it explain how findings in Hanabi might translate to real-world collaborative tasks. The link between Hanabi’s in-game interactions and broader applications remains unclear, and this omission leaves open questions about the broader relevance and applicability of the research.

- While the authors cite work by Hu and Sadigh as justification for their choice of Hanabi and as a baseline for comparison, this prior research does not fully support the tasks explored here. A central focus of Hu and Sadigh’s work was on human-AI coordination, which this paper does not directly address. This gap makes the comparison less relevant, as the primary objectives differ significantly.

- The authors critique Hu and Sadigh's approach for requiring an explicit specification of the expected behavior of the learning agent. However, the proposed R3D2 method also requires specific environment settings, which could present a similar or greater burden. The need to configure a consistent environment specification might, in some ways, be less useful than specifying interaction strategies, as discussed in later sections.

- A significant oversight in this paper is the lack of reference to the extensive body of literature on unsupervised environment design in both single-agent and multi-agent settings [1,2]. This research area directly addresses the problem of adaptability and generalization in agent training, which the authors claim to investigate. For the paper’s contributions to be meaningful within the field, it is essential for the authors to engage with this related work and to include comparative empirical evaluations.

- While creating a text-based version of Hanabi contributes to the approach, it is unclear how this addresses the problem of adaptability that the authors set out to solve. Converting the environment to a text-based form, and leveraging symmetries within Hanabi, simplifies many of the original environment’s complexities. As a result, adaptability may not be fully tested here since the text-based version removes many elements that would typically require generalization, making it a fundamentally easier learning problem.
- The approach also necessitates adapting the replay buffer to handle a variable number of players, which could be a limiting factor. Although this modification works here because Hanabi caps players at five, it may not scale effectively for environments requiring more flexibility or involving larger numbers of agents.

- The empirical evaluation of collaboration with other agents is limited to two-player settings, as the other agents require training specific to each number of players. This narrow scope restricts the evaluation, as it remains unclear how these methods would perform in settings with more players. Additionally, because the baseline agents are not trained using a text-based representation, this could lead to an unfair comparison between approaches.

- The authors indicate that all agents were trained using identical hyperparameters (line 374-375), despite the agents having different architectures and representations. Given these differences, it is crucial to tune hyperparameters individually for each agent to ensure a fair comparison, as uniform parameters may disadvantage some agents more than others.

- The explanation given for the relatively low performance of OP (line 410) lacks clarity, and the observed performance of OBL seems promising in comparison to R3D2. However, the current experimental setup does not provide sufficient data to substantiate why R3D2 should be chosen over OBL or how these results might generalize across settings.

- The passage on lines 422-425 presents an unclear and convoluted explanation of the comparison between methods, and the reported marginal improvements in R3D2’s performance relative to other settings are not fully justified within the results.

- The insights gained from the results presented in Figure 6 are unclear. Given that R3D2 was trained across all game environments, it is unsurprising that it performs adequately across settings. Without a clearer interpretation, the results lack novelty, as the outcomes align with the fact that R3D2 has been exposed to all environments during training.

[1] Emergent Complexity and Zero-shot Transfer via Unsupervised Environment Design, Dennis et. al. NeurIPS 2020

[2] MAESTRO: Open-Ended Environment Design for Multi-Agent Reinforcement Learning, Samvelyan et. al. ICLR 2023

**Questions:**

- Could the authors clarify their rationale for selecting Hanabi as a testbed for studying adaptability to new settings and new co-players? Specifically, in what ways does Hanabi reflect the complexities of real-world collaborative tasks, as mentioned in the introduction?

- How does the proposed approach relate to the existing literature on Unsupervised Environment Design, particularly for multi-agent reinforcement learning? Were there considerations given to this body of work in framing the method, and if so, how does the R3D2 approach build on or diverge from these methods?

- How scalable is the R3D2 approach with respect to increasing the number of players? Given that the current implementation handles a maximum of five players, are there foreseeable limitations if applied to environments with a larger number of agents?

- On line 205, it is mentioned that the text representation “includes the knowledge of the player’s own hand.” Could the authors clarify if this refers to the player’s hand as revealed by others through clues up to the current time point?

- Are the baseline algorithms (such as OP and OBL) trained on the text-based version of the Hanabi environment or on the original bitstring encoding? If not on the text-based version, could the authors address how this may impact the fairness of the comparison?

- Given the different architectures and representations used for each baseline, why were all agents trained with identical hyperparameters (as noted on line 374-375)? Could the authors discuss whether hyperparameter tuning was considered for each model to ensure the most accurate and fair evaluation of baseline performance?

- The explanation for OP’s lower performance on line 410 is somewhat vague. Could the authors provide additional insights or analysis on why OP underperformed relative to R3D2, as well as a more detailed comparison between R3D2 and OBL?

---

> ### Author Response · Authors · 2024-11-20
>
> We would like to thank you for your extensive and in-depth review of our work. We appreciate you find our focus addresses a critical gap in MARL research and that, within Hanabi, we address this gap by achieving effective transfer of strategies across different game settings.
>
> We would like to address your comments as follows:
>
>
> > in what ways does Hanabi reflect the complexities of real-world collaborative tasks, as mentioned in the introduction? The paper does not sufficiently establish why Hanabi is an appropriate testbed for studying adaptability to new settings and collaborators
>
> Thank you for this comment, which has been raised by other reviewers as well. We provide a detailed answer to these questions in the general comments **[Why we focused on Hanabi?]** and **[Is R3D2 architecture task agnostic?]**. To summarize, we focus on Hanabi mainly because:
>
> Hanabi has been shown to correlate with improved human-AI coordination (Hu et al. (ICML 2020), Hu et al. (ICML 2023)) and demonstrated that strategies learned in Hanabi can transfer to human-AI collaboration scenarios.
> We can study Zero-shot adaptation to novel partners and transfer across player settings,  where the goal is to have a team of agents that works well together (Cui et al. (NeurIPS 2022))
> The Hanabi game incorporates key challenges found in real-world multi-agent scenarios: partial observability, communication constraints, and the need for long-term planning Bard et al. (2020).
>
> While we demonstrate our approach on Hanabi, our core technical contributions - text-based representation for better transfer, architecture for dynamic action/state spaces, and variable-player learning - are domain-agnostic advances that could benefit MARL applications. Previous works like Foerster et al. (ICML 2019), Hu et al. (ICML 2020), Cui et al. (NeurIPS 2022), Hu et al. (ICML 2021) have successfully used Hanabi to develop fundamental MARL concepts that have influenced broader cooperative AI research.
>
> > While the authors cite work by Hu and Sadigh as justification for their choice of Hanabi and as a baseline for comparison, this prior research does not fully support the tasks explored here
>
> We cite Hu & Sadigh (2023) primarily to acknowledge another approach that incorporates language models in Hanabi for improving cross-play performance, though in a different context (human-AI coordination). While their work includes human evaluation, they also report self-play and intra-algorithmic cross-play scores that provide relevant benchmarks. Although our frameworks address different aspects of coordination (human-AI vs zero-shot), including their work in our literature review provides important context about language-based approaches in Hanabi. However, we do not use their method as a baseline since their objective (adapting to human-specified strategies) differs fundamentally from ours (achieving zero-shot coordination through representation learning). We updated the related work section to make the distinction more clear.
>
> > adaptability may not be fully tested here since the text-based version removes many elements that would typically require generalization, making it a fundamentally easier learning problem.
>
> Text is instrumental in enabling generalization and transfer, which is a central focus of our paper. Notably, we are the first to propose a learning agent capable of simultaneously playing across all Hanabi settings while generalizing zero-shot to novel partners and game configurations. This capability has not been demonstrated by any previous SOTA or baseline methods.
>
> We also performed additional ablation studies, we created an intermediate baseline called R2D2-text (R2D2 backbone, recieving states as text) to understand the role of language. Our results demonstrate that merely converting states to text is insufficient for full generalization - the R3D2 architecture is crucial for adaptation to novel partners and game settings. This shows that while text representation simplifies some aspects of generalization, the combination of appropriate representation and architecture is necessary for robust performance. For reference,  in section 6.2 and A.7 ,
>
>
> > The approach also necessitates adapting the replay buffer to handle a variable number of players, which could be a limiting factor.
>
> We fully agree that, to handle a variable number of players, we had to pad sequences, a standard practice in deep learning. The original R2D2 implementation for Hanabi (Hu et al., 2021c) also pads episodes to length 80 since episode lengths naturally vary.

---

> ### Author Response · Authors · 2024-11-20
>
> > How scalable is the R3D2 approach with respect to increasing the number of players? Given that the current implementation handles a maximum of five players, are there foreseeable limitations if applied to environments with a larger number of agents?
>
> The size of the state space increases slightly to capture the extra hand of the new players, and does not require any architectural changes. In our case, we use a BERT layer to process the textual observation. The maximum context length for BERT is 512 tokens. Our 2-player observation requires 96 tokens. This increases linearly with around 33 additional tokens per extra player, for a maximum of 12 additional players in Hanabi.
>
> In general, our method is thus limited by the maximum context length. This could be increased with larger language models, although it would considerably increase training in terms of walltime. Still, the observation-size also increases for the bitstring encoding. Growing observation-size proportional to the number of agents is a general concern for MARL agents.
> >  lines 422-425 presents an unclear and convoluted explanation of the comparison between methods
>
> We apologize if our explanation seemed convoluted. What we wanted to say was that R3D2 has a better cross-play capacity than OBL since, regardless of which algorithm it is paired with, R3D2 has a higher score than OBL. That is, R3D2 and OP cooperate better together than OBL and OP. This is also true for R3D2 and IQL compared to OBL and IQL. We have rewritten these lines in the updated version of the paper for better understanding and clarity (line 495-500).
>
> > Given that R3D2 was trained across all game environments, it is unsurprising that it performs adequately across settings.
>
> We respectfully disagree. Developing an architecture that can handle both dynamic action and state spaces requires careful consideration of representation learning and architectural choices. Our work is the first to demonstrate that a single model can successfully master all Hanabi game settings, a non-trivial technical achievement that opens new possibilities for flexible multi-agent systems. Moreover, We have 2 setups in R3D2,
> - **[Zero-shot Transfer]** R3D2-Single setting (R3D2-S), can handle dynamic action and state spaces which was not possible in the baseline approaches. We show that an R3D2-S agent trained in a 2-player setting can play in a 3,4,5-player setting reasonably well without additional training or architectural modification (Figure 4).
> - **[Multi-Task MARL]** R3D2-Multi setting, (R3D2-M) makes it possible to train one single model to play all the Hanabi games simultaneously using the same amount of training budget. With the baselines, it is standard to train individual models to play in different settings, which doesn’t resonate well with real-world scenarios. For example, a human player can play 2, 3, 4, 5 player games. R3D2-M is the first generalist Hanabi agent, since it can play on all game settings at the same time, but most of our cross-play results focus on R3D2-S.
>
> More detailed comments on the concern on performance are addressed in the general response section **[About R3D2’s performance]**
>
> > The authors make several broad claims in the introduction about multi-agent reinforcement learning (MARL) and the adaptability of agents, but these claims lack supporting evidence or citations
> We grounded our statements by adding supporting citations in our updated version of the paper.
>
> > How does the proposed approach relate to the existing literature on Unsupervised Environment Design
>
> We thank the reviewer for this question. In our work, we proposed the first agent capable of playing all Hanabi settings simultaneously while also generalizing zero-shot to novel partners and game configurations. We have expanded our related work section to better position our approach in relation to the UED literature. While works like Dennis et al. (2020) and Samvelyan et al. (2023) tackle transfer learning by automatically generating training environments of increasing complexity, our approach takes a fundamentally different direction. We reformulate the original environment using language as a unified representation that enables generalization across different game configurations. The key distinction is that UED approaches seek to create diverse training scenarios to help agents develop robust policies, while we maintain the original environment but change how agents perceive and interact with it through text-based representations. Our results suggest that appropriate representation choice can achieve strong generalization without requiring environment modification, though combining both approaches would be a really interesting avenue for future work.
>
> > Could the authors clarify if this refers to the player’s hand as revealed by others through clues up to the current time point?
>
> Yes, exactly!

---

> > ### Author Response · Authors · 2024-11-20
> >
> > > Could the authors discuss whether hyperparameter tuning was considered for each model to ensure the most accurate and fair evaluation of baseline performance?
> >
> > For baseline agents (IQL, OP, and OBL), we deliberately used the exact hyperparameter configurations reported in the OBL paper (Hu et al., 2021), which have been extensively tuned and validated by the community. This choice ensures we compare against these methods at their established best performance and maintains consistency with published results.
> >
> > Importantly, for all other architectural and training parameters for our approach (learning rate, network sizes, replay buffer configuration, etc.), we used the same values as the baselines. The fact that R3D2 achieves strong performance without extensive hyperparameter tuning of its RL components demonstrates the robustness of our approach.
> >
> > For R3D2, we focused our tuning efforts primarily on the language model component, exploring:
> >
> > - The impact of using pre-trained versus randomly initialized weights
> > - Different update frequencies for the language model during training
> > - Various small language model architectures (BERT, DistilBERT, TinyBERT)
> >
> > We refer to the general comment for detailed results on the ablation **[Role of language and language model].**
> >
> > > The explanation for OP’s lower performance on line 410 is somewhat vague. Could the authors provide additional insights or analysis on why OP underperformed relative to R3D2, as well as a more detailed comparison between R3D2 and OBL?
> >
> > Thank you for your insightful comment. We have incorporated this feedback into the revised version of our paper to provide greater clarity on the OP’s lower performance, specifically addressed in line 483. To support our explanation, we refer to the official results from OP paper (Hu et al. (2020)) in which inconsistent performance behavior was also observed when integrating OP with IQL.
> >
> > While OBL is a significant step in achieving ZSC, our goal was to design a simple, scalable and robust agent with no architectural changes for different player settings which can coordinate with novel partners and game settings only using self-play. The fact that R3D2 achieves higher inter-XP scores with OBL agents also shows that R3D2 learns more human-like strategies since OBL has been shown to coordinate well with humans. However, we leave that study as a future work on benchmarking with human evaluations. Moreover, our approach is orthogonal to existing advances in zero-shot coordination - methods like Other-Play, trajectory diversity, or Off-belief Learning could potentially be combined with our text-based representation to achieve even stronger generalization performance.
> >
> > **We would like to thank you again for this in-depth review. We hope that our answers show our work is well-grounded, with extensive analysis concerning the architecture, and its generalizability. If we have adequately addressed your concerns, we kindly ask for your consideration in enhancing the score you've allotted to our submission. Otherwise, we would be glad to answer any additional questions you may have.**

---

> > > ### Author Response · Authors · 2024-11-28
> > >
> > > Dear Reviewer ubVW,
> > >
> > > We would like to thank you again for your valuable feedback. The constructive discussion with all the reviewers has helped us to substantially improve the paper. We would greatly appreciate your feedback on our responses. Please let us know if any points need further clarification.
> > >
> > > Best regards,
> > >
> > > The Authors

---

> > > > ### Comment · Reviewer_ubVW · 2024-12-02
> > > > **Thank you for detailed response**
> > > >
> > > > Dear Authors -- I commend your efforts in responding to all my questions in great detail, especially in light of the amount of reviews you received, your time and patience is highly appreciated.
> > > > After carefully reading your responses to my and other reviewer's comments, your global responses and your revision to the paper (which are also appreciated), I would like to maintain my score as I still have reservations for the work in its current form.
> > > >
> > > > - While the authors have made efforts to motivate the use of Hanabi, it is not convincing when it comes to the general applicability of the approach beyond Hanabi for other MARL applications.In fact, I recommend that the authors do not use claims such as line 91-94 in the updated version of the paper because those claims are unsubstantiated inits current form of the work and also very difficult to directly imply. I believe that rather than trying to show why developing this approach on Hanabi can lead to generalized ability beyond the environment, it might help to go to back to the initiation of the project and discuss more comprehensively about  what led to the choice of Hanabi when thinking about this approach.
> > > >
> > > > - This is because currently choice of Hanabi appears to be a limiting factor for it to be applicable to any other domain e.g.
> > > > total number of players, scalability issues as described in initial review, Hanabi being game highly amenable to textual representation (again a current limitation that authors acknowledge in their responses) and others.
> > > >
> > > > - The authors rely on the previous works to motivate use of Hanabi, but I am sure the authors appreciate that most of those citations are from pre-LLM era except Hu et. al. (which has a different focus as authors state and I agree). While the authors provide current performance on LLMs which I listed in my strengths, it is not clear how the use of the current architecture age with the advancements in LLMs itself.
> > > >
> > > > - Further, the authors state that, "Developing an architecture that can handle both dynamic action and state spaces requires careful consideration of representation learning and architectural choices. Our work is the first to demonstrate that a single model can successfully master all Hanabi game settings, a non-trivial technical achievement that opens new possibilities for flexible multi-agent systems." While I agree with this comment in spirit, this is precisely why choice of Hanabi is a major limiting factor for the approach authors are trying to propose. The issue is that changing the domain or environment may require much more complicated exercise for coming up with an architecture that can handle both dynamic action and state spaces and careful representation learning i.e. the current architecture may not at all be useful in other domain (it may be but it is hard to say).
> > > >
> > > > - Finally, I appreciate authors reporting more experiments in the Appendix to consider the effect of text vs effect of architectural choices. However, this does not address my original concern fully/ probably the authors missed part of my concern but don't blame them for that. Before stating my point, I want to say that the experiments are more than one can ask during the rebuttal period, especially given so many reviews and I do appreciate these efforts so not asking for any more experiments. But my point was that text makes the environment much easier for agents, already solving some of the generalization related changes for the agents. I never meant to imply that all the performance is coming just from the text conversion. I see the architectural changes as necessary additions to support variable players effectively and so adding those architectural choices (again I am worried they might only work for Hanabi) is fine. But the part of my concern that was missed was about other baselines which never had text at their disposal. To be precise, between R3D2 and baselines (other than R2D2), there are two changes, availability of text based representation and different architecture. and so comparison with them seems a bit unconvincing. Finally, the appendix results are only for 2 players and as I mentioned, I cannot expect more but at same time, its futile exercise as it is hard to see if they will hold for more players.
> > > >
> > > > In sum, the paper has some nice experiments and analysis on generalization in text based Hanabi environment but any broader impact beyond that to general MARL community seems a stretch at this point.
> > > >
> > > > I'm sorry in case this paper doesn't go through because I understand the amount of efforts that the authors would have put in this work and I hope that my comments help the authors to give some food for thought on improving over this work either in the in the final version of this paper or in the next version.

---

> > > > > ### Author Response · Authors · 2024-12-02
> > > > >
> > > > > > it is not convincing when it comes to the general applicability of the approach beyond Hanabi for other MARL applications
> > > > > > the current architecture may not at all be useful in other domain
> > > > >
> > > > > We appreciate the reviewer's concern about the generalizability claims. While we acknowledge that our approach is specifically developed and tested within the Hanabi domain, we align with Reviewer 5SDa's perspective that "solving Hanabi is, in my opinion, a sufficient goal for a paper. Hanabi is a tough MARL benchmark and solving it is a worthy goal even without immediate applicability to real world problems, similar to other RL problems that were the center of famous papers (Atari, Go, NetHack etc.)."
> > > > >
> > > > > That said, we see promising indicators for potential broader applicability of our approach. At its core, R3D2 builds upon self-play, which has been successfully applied across various multi-agent cooperation problems [1,2]. Our architecture combines this proven foundation with two key innovations that we believe could transfer to other domains: textual interactions and a dynamic network architecture for actions based on DRRN architecture. Previous work has demonstrated that language can improve transfer capabilities [3,4], and DRRN has shown strong generalization performance in multi-task settings, as evidenced by its superior performance in ScienceWorld [5], an environment encompassing 30 different tasks with numerous variations.
> > > > >
> > > > > To demonstrate the robustness of our approach even within Hanabi, we have expanded our evaluations to include 3, 4, and 5-player games (detailed in Appendix A.6), showing that R3D2's performance advantages scale with increasing numbers of agents.
> > > > >
> > > > >
> > > > > However, we acknowledge that further research would be needed to validate these potential benefits in other domains. In the revised manuscript, we will refocus our presentation to emphasize our concrete achievements within the Hanabi domain, removing broader generalization claims that extend beyond our demonstrated results.
> > > > >
> > > > > [1] Berner et al. (2019), Dota 2 with large scale deep reinforcement learning. arXiv preprint arXiv:1912.06680.
> > > > >
> > > > > [2] Lin et al. (2023), Tizero: Mastering multi-agent football with curriculum learning and self-play. arXiv preprint arXiv:2302.07515.
> > > > >
> > > > > [3] Radford et al.,  Language models are unsupervised multitask learners. OpenAI Blog, 1(8):9, 2019b.
> > > > >
> > > > > [4] Brown et al., Language models are few-shot learners. Advances in neural information processing systems, 33:1877–1901, 2020.
> > > > >
> > > > > [5] Wang et al. ScienceWorld: Is your Agent Smarter than a 5th Grader? In Y. Goldberg, Z. Kozareva, & Y. Zhang (Eds.), Proceedings of the 2022 Conference on Empirical Methods in Natural Language Processing (pp. 11279–11298)
> > > > >
> > > > > > between R3D2 and baselines (other than R2D2), there are two changes, availability of text based representation and different architecture. and so comparison with them seems a bit unconvincing.
> > > > >
> > > > > Indeed we previously did not have an intermediate baseline with only one change. However, we have worked hard to improve that aspect, by providing **additional ablation experiments that introduce R2D2-text (Same architecture as R2D2 with only text as input)**, which show that, **while text does offer some help for transfer, handling dynamic action spaces is critical for successful transfer to novel partners** (Please checkout our updated Figure 4 in Section 6). This new ablation study provides a clearer picture of the individual contributions of each component.
> > > > >
> > > > > > The authors rely on the previous works to motivate use of Hanabi, but I am sure the authors appreciate that most of those citations are from pre-LLM era except Hu et. al. (which has a different focus as authors state and I agree). While the authors provide current performance on LLMs which I listed in my strengths, it is not clear how the use of the current architecture age with the advancements in LLMs itself.
> > > > >
> > > > > The modular nature of our R3D2 architecture actually positions us well with respect to LLM advancements. R2D2-text shows that using textual representations improves cross-play performance. As language models continue to improve their ability to create better representations, we can replace our current language models with newer ones. This is because,  as we discussed in **[3. Is the R3D2 architecture task agnostic?]**, our architecture's language model component is designed to be replaceable. And, while our current citations do indeed draw heavily from pre-LLM era work, this reflects the foundational nature of these papers in establishing Hanabi as a benchmark, rather than a limitation of our approach.

---

> > > > > > ### Author Response · Authors · 2024-12-02
> > > > > >
> > > > > > > appendix results are only for 2 players and as I mentioned, I cannot expect more but at same time, its futile exercise as it is hard to see if they will hold for more players.
> > > > > >
> > > > > > We respectfully disagree with the characterization of our results as limited to 2-player settings. As demonstrated in Figure 12 of Appendix A.6, we have conducted extensive experiments across 3-, 4-, and 5-player configurations, presenting both Self-Play (SP) and intra-Cross-Play (intra-XP) scores. To our knowledge, this represents one of the most comprehensive evaluations in Hanabi research, specifically designed to validate that our method's performance is not limited to any single game configuration. The results demonstrate consistent strong performance across all player counts, suggesting robust generalization capabilities within the Hanabi domain.

---

### Official Review · Reviewer_uDKb · 2024-11-12

**Soundness:** 3
**Presentation:** 3
**Contribution:** 3
**Rating:** 6
**Confidence:** 3

**Summary:**

This paper addresses a fundamental challenge in multi-agent reinforcement learning (RL): the difficulty of achieving robust performance across environments beyond those the agents were specifically trained on. To tackle these issues, the authors propose and evaluate their approach in a well-established environment for multi-agent RL, Hanabi, which is commonly used to test cooperation in complex multi-agent settings. The paper makes two primary contributions. First, it reformulates the task using textual representations, motivated by evidence that language can improve transferability. By representing the Hanabi game through text, the authors aim to create agents that are more adaptable and capable of generalizing beyond their training environment. Second, they introduce a novel neural network architecture that combines a language model with the Deep Recurrent Relevance Q-network (DRRN). The results presented indicate that the proposed method achieves high intra-algorithm cross-play scores when trained in a self-play setting, underscoring its capacity for adaptability and teamwork. Additionally, the method performs exceptionally well in inter-algorithm play, highlighting its capability to cooperate with agents utilizing different algorithms.

**Strengths:**

1. One of the primary strengths of this work is its success in achieving effective transfer learning in a zero-shot setting. The proposed method shows adaptability across varying scenarios without requiring additional retraining, which addresses a significant limitation in existing multi-agent RL models.

2. Through careful evaluation, the authors demonstrate that their approach outperforms existing methods in inter-algorithm play, achieving high scores in cross-play with agents using different algorithms.

3. By representing the Hanabi game through text, this work leverages advances in natural language processing (NLP) and large language models (LLMs), marking a novel direction in multi-agent RL. Their approach allows the potential for more seamless integration with future advancements in language models.

**Weaknesses:**

1. The reliance on text-based representation, while novel, may face significant challenges in more complex or non-linguistic domains. For instance, domains where numeric data dominates—such as algorithmic trading, where key data points include stock prices and volumes, or robot-taxi applications, where visual and sensory data are crucial—may not naturally fit into a textual framework. An extension or discussion on the applicability limits of textual representation in various domains would provide valuable context and guide future research on adapting the method to broader applications.

2. The self-play performance is relatively poor. Insights into possible contributing factors, such as model structure or training limitations, would be useful. Additionally, outlining potential directions for improving self-play performance, whether through modifications to the architecture, training process, or environment setup, would provide actionable guidance for future work.

3. There is insufficient discussion on the choice of neural network architecture, particularly regarding the component where the element-wise dot product is applied between the embedded observations and actions. Similar ideas have been explored in the literature, such as in https://proceedings.mlr.press/v202/ma23e/ma23e.pdf. The authors may consider incorporating insights from existing explanations in the literature to better articulate the advantages and underlying mechanisms of this architectural choice.

**Questions:**

1. Could the authors clarify why the focus is primarily on self-play during the training phase? While it is impressive that the proposed method achieves high cross-play scores after self-play training, it seems limiting not to explore other training strategies, especially when inter-algorithm play is a key evaluation goal.

2. Given the novel combination of language model and DRRN components, was there a reason for not experimenting with a wider variety of neural network architectures? Exploring different architectures might reveal other configurations that could improve performance, especially in self-play or more complex settings. Were alternative architectures considered during development, and if so, what were the outcomes? Additional details about architectural choices and trade-offs would be helpful.

---

> ### Author Response · Authors · 2024-11-20
>
> Thank you for your insightful comments on our paper, and for recognizing one of the main strengths of our paper: that R3D2 shows adaptability across varying scenarios without requiring additional retraining, which addresses a significant limitation in existing multi-agent RL models.
>
> Thank you also for raising valid concerns and potential limitations of our work, which we are eager to address here.
>
> > An extension or discussion on the applicability limits of textual representation in various domains would provide valuable context.
>
> We acknowledge that the applicability of text representation to some domains might not be as straightforward as Hanabi. We address this more in detail in the general comment **[Is R3D2 architecture task agnostic?]**. To summarize, we updated Section 7 to highlight this limitation, pointed to interesting avenues for future work, and provided examples where specialized tokenizers could be used to alleviate the limitations of text.
>
> > The self-play performance is relatively poor
>
> We acknowledge your concern regarding R3D2’s performance and took it very seriously. To address this, we resumed training R3D2, which had previously undergone fewer training iterations due to computational constraints. As detailed in the general response **[About R3D2’s performance]**, this additional training led to significant improvements in self-play (22.23 → 23.34) and intra-XP scores (19.66 → 23.23).
>
> > Could the authors clarify why the focus is primarily on self-play during the training phase?
>
> Thank you for this question. As mentioned in lines 76-82 in the paper’s introduction section, we **deliberately focused on self-play to demonstrate a key insight: the choice of representation can fundamentally impact the robustness of learned strategies, even with simple training approaches**. While previous work like Hu et al. (ICML 2020) and Cui et al. (NeurIPS 2021) developed sophisticated methods to achieve zero-shot coordination, we show that self-play - typically criticized for learning brittle, specialized conventions - can actually learn robust, transferable strategies when combined with appropriate representations.
> Our results demonstrate that by using text-based representation, self-play agents can achieve strong cross-play performance without requiring complex training methods. This is significant because **self-play is simpler, more scalable, and computationally efficient compared to methods that require population-based training or hierarchical policies**. Moreover, our approach is orthogonal to existing advances in zero-shot coordination - methods like Other-Play, trajectory diversity, or Off-belief Learning could potentially be combined with our text-based representation to achieve even stronger generalization performance.
> This finding opens up new research directions for improving multi-agent coordination by focusing on representation learning alongside algorithmic innovations.
>
> > Were alternative architectures considered?
>
> Thank you for pointing us to (Ma et al., 2023). We are using an architecture akin to Ma et al.’s Self-Attention with Action as Inputs approach (Fig3-bottom), while our baselines are based on the Fig3-top (R2D2). Fig3-bottom is the best performing combination observed by Ma et al. (2023). With DRRN, this further confirms our architectural choice. We also added this discussion in Section 4.2 of our paper’s revision.
>
> Furthermore, we performed a number of ablations and experiments on the integration of different language model architectures (Section 5, Appendix A.4), which we detail in the general comment on **[Role of language and language model]**.
>
>
> **To conclude, we are grateful for your thoughtful comments and insightful questions, which helped us further refine our work and address key aspects of R3D2’s contributions and limitations. We hope these updates effectively address your concerns and demonstrate the significance of our approach. If our revisions meet your expectations, we kindly request you to reconsider the score you have assigned. Should you have further questions or require additional clarifications, we are more than happy to provide them.**

---

### Official Review · Reviewer_YgHj · 2024-11-13

**Soundness:** 3
**Presentation:** 3
**Contribution:** 3
**Rating:** 6
**Confidence:** 4

**Summary:**

This paper proposes a new method to train an agent to be able to play the game of Hanabi with the ability to generalize to all game settings and to be able to play with other agents in a zero-shot coordination capability. They do this by reformulating the task as a text task by converting current state and an action and play history into a text representation, embedding it using a language model and then feeding these embeddings into a Deep Recurrent Relevance Q-Network (DRRN) trained in a distributed way with a shared replay buffer. Their architecture called R3D2 uses R2D2 (Recurrent Replay Distributed Deep Q Network) as it’s foundation.

The biggest claim of this work is that no other work before their work has been able to generalize Hanabi agents across game settings.

**Strengths:**

1.	The paper is motivated well and written clearly, with a sufficient explanation of the game and the existing literature to bring the reader upto speed
2.	The evaluation and experiments are thorough, ablating and justifying all of the pieces. They start with first demonstrating that LM only is not sufficient to play the game. Then they show comparisons against 3 existing approaches.

**Weaknesses:**

1. The work is Hanabi-centric. It may be possible to apply this same work to other card games as it's mostly the text representation that would change
2. Even with its Hanabi focus, I am unable to see the claims being made in the results section. R3D2 scores lower on Self-play and intra-XP but higher on inter-XP. The grid in Fig 5, also does not have massive increase in scores against R3D2

**Questions:**

1. Some parts of the setup were a little confusing. eg: “All baselines use R2D2 as a basis” vs “we utilized OP and OBL checkpoints from the original paper which focused exclusively on the 2-player setting”. Were the original checkpoints foundationally R2D2?
2. Is a jump of 3-5 points a big jump in the game of Hanabi, regarding Table 5?

---

> ### Author Response · Authors · 2024-11-20
>
> Thank you for your insightful comments and feedback. We are glad that you found our paper well written, and that you found the experiments thorough. Showing that text only is not sufficient to learn to cooperate is an important aspect of our work, which we appreciate you noted down.
>
> You made some very valid points in the review, which we aim to address appropriately below.
>
> > work is Hanabi-centric
>
> We address this question in a general response **[Why we focused on Hanabi?]**, but to summarize, we indeed focused on Hanabi because:
>
> 1. Hanabi has been shown to correlate with improved human-AI coordination (Hu et al. (ICML 2020), Hu et al. (ICML 2023)) and demonstrated that strategies learned in Hanabi can transfer to human-AI collaboration scenarios.
> 2. We can study Zero-shot adaptation to novel partners and transfer across player settings,  where the goal is to have a team of agents that works well together (Cui et al. (NeurIPS 2022))
> 3. Finally, the Hanabi game incorporates key challenges found in real-world multi-agent scenarios: partial observability, communication constraints, and the need for long-term planning (Bard et al. (2020)).
>
> > R3D2 scores lower on Self-play and intra-XP but higher on inter-XP. The grid in Fig 5, also does not have massive increase in scores against R3D3.
> > Is a jump of 3-5 points a big jump in the game of Hanabi, regarding Table 5?
>
> We understand your concern about R3D2’s performance, and took this comment very seriously. We resumed training of R3D2, which had less training iterations due to computational constraints. We expand upon this in the general response **[About R3D2’s performance]** but, concretely, this resulted in a **significant improvement on selfplay (22.23 -> 23.34)  and intra-XP scores (19.66 -> 23.23)**.
>
> A 3-5 point improvement is indeed significant in Hanabi. The game has a maximum possible score of 25 points, and improvements of even 1-2 points at higher performance levels are considered meaningful in the literature. For context, prior works like Foerster et al. (ICML 2019), Hu et al. (ICML 2020) report improvements in the range of 1-3 points as substantial advances. Additionally, the difficulty of improving scores increases non-linearly as performance approaches the ceiling, making 3-5 point gains even more notable.
>
> > “All baselines use R2D2 as a basis” vs “we utilized OP and OBL checkpoints from the original paper which focused exclusively on the 2-player setting”. Were the original checkpoints foundationally R2D2?
>
> Yes, all the baselines are foundationally R2D2.
>
> **To conclude, we sincerely thank you for your valuable feedback and insightful comments. We have taken your concerns seriously and have worked diligently to address them through detailed explanations and additional training of R3D2, leading to significant improvements in performance. If you find that our revisions and explanations have sufficiently resolved your concerns, we kindly request your consideration in revisiting the score you have assigned. Should you have any further questions or need additional clarifications, we would be more than happy to address them.**

---

> > ### Comment · Reviewer_YgHj · 2024-11-24
> >
> > Thank you for your response, I will be reading through rest of the discussions and reconsidering my score

---

> > ### Comment · Reviewer_YgHj · 2024-11-25
> >
> > It is laudable that the authors read through and responded to 12 reviews; resumed training of checkpoints and updated their related work section, appendix and expanded experiments to 3,4 players. I am raising my score to a 6 and I urge the other reviewers to make time to read through the discussions and updates.

---

### Official Review · Reviewer_wJ2W · 2024-11-14

**Soundness:** 2
**Presentation:** 3
**Contribution:** 2
**Rating:** 5
**Confidence:** 3

**Summary:**

The paper introduces a new approach to utilizing text generation to enable knowledge transfer in more than 2 agents in Hanabi tasks. The proposed method shows performance improvements with different LLMs in ZSC settings. Overall, the paper is well-written and shows some interesting inspirations.

**Strengths:**

- The paper is well-written
- The idea seems to be promising and novel
- The experiments are extensive

**Weaknesses:**

- the text template is not clear enough. What information is included in the text template? Will the text be revealed to other agents? It seems that the template will record all "previous" information which is simply the history of the observations. Then why not use history instead of text recording?
- the author said R2D2 is a foundation of R3D2, but it does not compare to R2D2. Moreover, since R2D2 is not introduced in detail, it is hard to perceive the technical contribution of R3D2
- what's the intuition behind multiplying the state and action embeddings? Where do you integrate the text information? how do you make sure it can accommodate different numbers of agents and task settings?

**Questions:**

Besides the questions from the weakness. I have a few concerns about the text templates used in the paper:
- if the text is embedded and used by the Q-functions, does it mean the text is better than observation features? Is there any intuition behind this?
- is there any communication between the agents? In Hanabi tasks, the agents can utilize hint actions to communicate information implicitly. If the text is now used, would this shift to explicit communication and potentially risk leaking private information?

---

> ### Author Response · Authors · 2024-11-20
>
> Thank you for your feedback on our submission. We appreciate that you find the experiments extensive. Proposing inter-algorithmic and inter-setting cross-play is a key contribution of our work, which we are glad to have been highlighted.
>
> We have carefully considered the concerns and would like to address them as follows:
>
> > text template is not clear enough. What information is included in the text template? Will the text be revealed to other agents? It seems that the template will record all "previous" information which is simply the history of the observations. Then why not use history instead of text recording?
>
> The template contains the current game state information - including tokens, played cards, visible hands, discards, and hints - as shown in Figure 1. **Each agent only receives the information it would normally observe in the game, simply formatted as text**. We choose text representation for both states and actions because it provides a more consistent state and action space across different player counts and enables transfer, as explained in lines. However, as is the case for R2D2, R3D2 uses RNNs to estimate the history of state-action values. The only difference is that, for R3D2, the state- and action-embeddings that are passed to the RNN result from TinyBERT layers.
>
> We updated the figure caption in the revision to be more descriptive to the readers.
>
> > it does not compare to R2D2. Moreover, since R2D2 is not introduced in detail, it is hard to perceive the technical contribution of R3D2.
>
> In this work, R2D2 is basically Independent Q-Learning (IQL) (Tan, 1993; Tampuu et al., 2017). IQL is still frequently used as a baseline for Hanabi, as it serves as the foundation of many state-of-the-art MARL Hanabi algorithms. It shows strong performance with its training partners, having learned highly specialized conventions through self-play. R2D2 are still independent learners, but introduce the following changes to improve the learning process:
> - R2D2 uses RNNs to cope with partial observability settings,
> - R2D2 collects experiences from multiple environments in parallel, speeding up global walltime,
> - R2D2 includes many best practices of DQN, namely double DQN, a dueling architecture, and prioritized experience replay.
> We hope these explanations, which are also explained on lines 208 - 212 in the paper, clarify the R2D2 agents.
>
> Moreover, to better understand the role of R3D2’s different components, we ran additional ablation experiments (Section 6.2 and A.7).
>
> > intuition behind multiplying the state and action embeddings? Where do you integrate the text information? how do you make sure it can accommodate different numbers of agents and task settings?
>
> The choice of elementwise multiplication for combining state and action embeddings is well-motivated by prior work. Our approach builds directly on the Deep Reinforcement Relevance Network (DRRN) architecture (He et. al, ACL 2019), which demonstrated the effectiveness of this operation for text-based games. This multiplicative interaction has been widely adopted in both deep learning and reinforcement learning as an effective way to condition outputs on contextual inputs, as shown in various domains including audio synthesis (Oord et. al, 2016), visual reasoning (Perez et. al, 2017), and sequential decision-making (Kumar et. al, 2019).
> Text information is integrated through BERT layers that encode both observations and actions into embeddings (shown in Figure 2). These text encoders transform the game state and action descriptions into fixed-size vectors that capture the semantic meaning while being agnostic to the specific number of players.
>
> The architecture accommodates different numbers of agents and settings because:
> - The text template naturally extends to include additional players without structural changes
> - The action space remains consistent since all actions are encoded as text
> - We use zero-padding in the replay buffer to handle varying sequence lengths.
>
> > if the text is embedded and used by the Q-functions, does it mean the text is better than observation features? Is there any intuition behind this?
>
> This is an interesting question. Text is not better than observation features per se, as R2D2, which uses a vectorized representation of the environment, also reaches a high self-play score. However, **we argue that a textual representation, combined with a general-purpose language model, is better for generalization and transfer learning** (Radford et al., 2019b; Brown et al., 2020). To better understand the role of text, we ran more ablation experiments which show that, while a text representation provides some benefits for generalization, handling dynamic action spaces is equally crucial for successful transfer to novel partners. Please refer to **[Role of language and language models]** for the role of language and language models in learning generalizable policies.

---

> > ### Author Response · Authors · 2024-11-20
> >
> > > is there any communication between the agents? In Hanabi tasks, the agents can utilize hint actions to communicate information implicitly. If the text is now used, would this shift to explicit communication and potentially risk leaking private information?
> >
> > As per the game rules of Hanabi, agents cannot communicate explicitly. The hint actions are still implemented as part of the standard game mechanics - when an agent gives a hint, it becomes part of the observable game state in text form, just as it would in the original game format. Thus, **even though we use text, we maintain the same information constraints as the original Hanabi environment.** For example an agent playing the action  “Reveal +1 color R” would hint the player about red cards in their hand. Red cards from said players will then be marked as red in his observation, formatted as “knowledge about own hand: Unknown X, Red X, Red X, Unknown X, Unknown X.”.
> >
> > **To conclude, we sincerely appreciate the time and effort you invested in reviewing our submission. Your constructive comments and questions have provided valuable guidance in clarifying and enhancing our work. We hope our revisions effectively address your feedback and demonstrate the contributions and strengths of our approach. If you find the revisions satisfactory, we kindly request your consideration in revisiting the score you’ve assigned. However, should there be any remaining concerns, we would be glad to clarify further.**

---

> > > ### Comment · Reviewer_wJ2W · 2024-11-26
> > >
> > > Thanks for the response of the authors. After reading the comments, I would like to maintain my scores.

---

### Author Response · Authors · 2024-11-20
**General response**

We gratefully acknowledge all the reviewers for their insightful comments. We are happy that reviewers found our work **well-motivated** [Reviewers YgHj, Axwq, YKqN, 5SDa, HW8S, u5D6], **written clearly** [Reviewers YgHj, w4wV, k3Nj, wJ2W] and that, through **thorough evaluation and experiments** [Reviewers YgHj, uDKb, 2tFM, k3Nj, wJ2W], our method (R3D2) shows **effective transfer across game settings** [Reviewers uDKb, ubVW, u5D6]. We would also like to thank Reviewers YKqN and k3Nj for finding our **evaluation design – cross-play across game-settings and algorithms – of interest** (Reviewer k3Nj stating he is a massive fan).

We would like to highlight our paper's main contribution: we present the **first agent capable of both playing all Hanabi settings simultaneously and generalizing zero-shot to novel partners and game configurations achieving this through self-play**, without complex MARL methods demonstrates a task-agnostic approach to generalization that we believe is valuable to the MARL community. Real-world multi-agent environments require agents that can adapt to dynamic settings and collaborate with diverse partners seamlessly, and R3D2 is a path towards that goal.

With a total of 12 reviews, we have received an unusually large amount of feedback. We have addressed the comments and queries raised by all the reviewers in the latest revision of our paper.

In summary, the revised paper incorporates the following updates, all of which are highlighted in blue for the reviewers' ease of reference:

- Updating the related work to include the literature on transfer learning and generalization in RL (Sec 2), as requested by [ubVW, k3Nj, uDKb]
- Resumed the training of R3D2 (no algorithmic changes were made), so it matches OBL in terms of training iterations. This resulted in significant performance improvements in Self-play and Cross-play (Sec 6.3). [2tFM, YKqN, YgHj, uDKb, 5SDa, k3Nj, u5D6]
- Expanded experiments to 3,4,5 Player settings (Appendix A.6) [ubVW]
- Adding ablation experiments on different components of R3D2 (by introducing a new baseline called R2D2-text) and the role of language representation as requested by [ubVW, w4wV, wJ2W, Axwq] which showed text representation alone is not enough for generalization and handling dynamic action spaces is equally crucial for successful transfer to novel partners. (Sec 6.3, Appendix A.7)
- Incorporated writing feedback from [ubVW, 5SDa]
- A mention that all code and models will be made public upon acceptance. (Appendix A.7.3)


These revisions have been made to address the valuable feedback provided by the reviewers, and allowed to enhance the paper in terms of clarity, significance and depth.

Furthermore, we here address common concerns by reviewers regarding:

1. **About R3D2’s performance,**
2. **Why we focused on Hanabi,**
3. **How task agnostic R3D2 is,**
4. **The role of language and language models,**

before answering each reviewer individually.


### [1. About R3D2’s performance]

It is important to note that the primary goal of our work is not to surpass existing baselines in terms of self-play performance but to show that self-play training (and not specialized methods) can achieve high cross-play and generalization to novel partners and game settings—key aspects of cooperation in multi-agent scenarios. **To the best of our knowledge, this is the first study to investigate generalization across game settings in Hanabi while relying solely on self-play for training.**

Despite this, we resumed the training of R3D2 to match OBL’s training iterations, from 2000 to 3000 epochs (we initially only trained for 2000 epochs due to computational constraints) with no algorithmic changes. As a result, **we observe improved self-play and cross-play performances** (See section 6.2 and 6.3 for updated results). Therefore, the improvements compared to the baselines were due to undertraining our agent and not their capabilities to learn generalizable policies. Here is a short summary of our updated results for the 2-player setting:

**R3D2: SP 22.23 -> 23.34, Intra-XP: 19.66 -> 23.23**

Compared to the baselines, our agents achieve competitive performance in self-play, intra-XP, and inter-XP scenarios.

---

> ### Author Response · Authors · 2024-11-20
>
> ### [2. About the focus on Hanabi]
>
> Hanabi has emerged as a popular benchmark for studying collaborative AI systems, as highlighted by Bard et al. (2020) who established it as a new frontier for AI research. The game incorporates key challenges found in real-world multi-agent scenarios: partial observability, communication constraints, theory of mind reasoning, and the need for long-term planning. These properties make it particularly relevant for studying adaptability to new collaborators and settings.
> Importantly, success in Hanabi has been shown to correlate with improved human-AI coordination (Hu et al. (ICML 2020), Hu et al. (ICML 2023)) demonstrated that strategies learned in Hanabi can transfer to human-AI collaboration scenarios.
>
> While we demonstrate our approach on Hanabi, **our core technical contributions - text-based representation for better transfer, architecture for dynamic action/state spaces, and variable-player learning - are domain-agnostic advances that could benefit MARL applications**. Previous works like Foerster et al. (ICML 2019), Hu et al. (ICML 2020), Cui et al. (NeurIPS 2022), Hu et al. (ICML 2021) have successfully used Hanabi to develop fundamental MARL concepts that have influenced broader cooperative AI research.
>
> We clarified the importance of Hanabi as a benchmark in the revision of the paper (Section 1).
>
> ---
>
> ### [3. Is the R3D2 architecture task agnostic?]
> We convert the textual information to tokens and compute the embedding using language models, where the embeddings capture the semantic meaning while being agnostic to the task. (Radford et al., 2019b; Brown et al., 2020). Then, we multiply state and action embeddings of language model inspired by the existing literature Deep Reinforcement Relevance Network (DRRN, He et al., ACL 2019) architecture, which is a popular baseline till now for most of the text-based games. This operation allows us to capture the relevance between states and actions in a way that generalizes across different game settings since both are encoded in the same semantic space through the language model.
>
> **Given we are using embedding and language models in our architecture, it is robust and easily adaptable to other tasks involving texts**. However, we acknowledge that the applicability of text representation to some domains might not be as straightforward as Hanabi. For example, environments like continuous control or vision-based tasks might require domain-specific adaptations (we updated Section 7 to highlight this limitation). We acknowledge this is a limitation of our work, however, recent advances in language models are expanding the possibilities. For instance, Llama-2's specialized tokenizer demonstrates remarkable performance on numerical tasks by decomposing numbers into digit sequences (Touvron et. al, 2023), suggesting exciting opportunities for extending our approach to domains with structured numerical representations.
>
> ---
> ### [4. Role of language and language model]
>
> To understand the role of language representation we performed the following ablations:
>
> 1. What is the role of pre-trained weights? (Appendix A7.1)
> - we compared the language model with pre-trained and random weights. Based on the results on five seeds, pre-trained weights significantly enhance sample efficiency. (Figure 12a for reference)
> 2. What is the role of updating the language model? (Appendix A7.1)
> - We aim to understand the role of updating the language model by changing the frequency of updates. A clear trend emerges: reducing the update frequency drastically diminishes the model's potential to reach peak performance.  (Figure 12b for reference )
> 3. Does the performance come from the language representation or the architecture?
>
> - To isolate the contributions of our two key innovations - using language models for state representation and handling dynamic action spaces - we created an intermediate baseline called R2D2-text.
> - **Our experiments demonstrate that text representation alone provides limited benefits for generalization. We show that R3D2 outperforms R2D2-text, demonstrating that the dynamic network structure is essential for successful transfer to novel partners.**. The superior performance of R3D2 compared to R2D2 and R2D2-text in both across game-setting transfer (updated the results of 6.2)  and cross-play (Added section A7.2)

---

### Public Comment · ~Dung_Viet_Nguyen1 · 2025-02-05
**Total respect for the authors!**

Hi, I've been rooting for you since the start of the rebuttal, especially with that unusual number of reviewers (12). Congratulations on your acceptance!

---

### Meta-Review · Area_Chair_CZ9Z · 2024-12-19

**Metareview:**

This paper proposes R3D2, a multi-agent RL algorithm for training a generalist agent for the Hanabi game. Major contributions include reformulating Hanabi as a text-based game to learn BERT-style representations and a distributed Deep Recurrent Relevance Q-Network with a shared replay buffer, using Recurrent Replay Distributed Deep Q Network as a foundation. The proposed agent is able to generalize across different game settings (e.g., varying player counts) and coordinate with unseen partners, achieving competitive empirical results.

The major strength of the work is its empirical results for this difficult game. The evaluation protocol is also comprehensive, including introducing new metrics like inter-XP and intra-XP, providing valuable insights into zero-shot coordination. The results demonstrate that self-play-trained agents can achieve strong cross-play performance, even matching human-coordination-oriented baselines like OBL in some settings.

The major concern from the reviewers mostly lie in the relevance of Hanabi. Many reviewers raised good points about general applicability of the approach beyond Hanabi for other MARL applications. Although the authors maintain that extending their approach to more practical domains lies outside the current scope, I strongly encourage the authors to explore more meaningful applications beyond game settings in future work.

**Additional Comments On Reviewer Discussion:**

During the rebuttal period, the authors addressed significant concerns raised across reviews, leading to notable improvements in the paper. Key points raised included: performance concerns, which were addressed by resuming training and demonstrating improvements; the role of language, clarified through new ablation studies, showing that text representation aids transfer and dynamic action space handling; lack of qualitative analysis, which was partially addressed by explaining cross-play generalization patterns; and insufficient comparison to baselines, rectified by introducing the R2D2-Text baseline to assess text representation.

Reviewers appreciated the improved clarity, added experiments, and detailed responses, leading to score increases from multiple reviewers, including those who raised their evaluations to 6 or 8. Despite some lingering concerns about broader applicability (ubVW, w4wV), the authors demonstrated that R3D2 achieves significant technical contributions, offering the first generalist Hanabi agent capable of strong self-play and cross-play performance across all game settings using a simple, scalable approach. These strengths informed the recommendation to accept the paper.

---

### Decision · Program_Chairs · 2025-01-22

Accept (Poster)